# Cross-species evolution of a highly potent AAV variant for therapeutic gene transfer and genome editing

Trevor J. Gonzalez [1], Katherine E. Simon[2,3,8], Leo O. Blondel [2,8],
Marco M. Fanous[2], Angela L. Roger [4], Maribel Santiago Maysonet[5],
Garth W. Devlin[2], Timothy J. Smith[1], Daniel K. Oh[2], L. Patrick Havlik[2],
Ruth M. Castellanos Rivera[5], Jorge A. Piedrahita[3], Mai K. ElMallah[4],
Charles A. Gersbach [6,7] & Aravind Asokan [1,2,6,7] ✉

Recombinant adeno-associated viral (AAV) vectors are a promising gene delivery platform, but ongoing clinical trials continue to highlight a relatively narrow therapeutic window. Effective clinical translation is confounded, at least in part, by differences in AAV biology across animal species. Here, we tackle this challenge by sequentially evolving AAV capsid libraries in mice, pigs and macaques. We discover a highly potent, cross-species compatible variant (AAV.cc47) that shows improved attributes benchmarked against AAV serotype 9 as evidenced by robust reporter and therapeutic gene expression, Cre recombination and CRISPR genome editing in normal and diseased mouse models. Enhanced transduction efficiency of AAV.cc47 vectors is further corroborated in macaques and pigs, providing a strong rationale for potential clinical translation into human gene therapies. We envision that ccAAV vectors may not only improve predictive modeling in preclinical studies, but also clinical translatability by broadening the therapeutic window of AAV based gene therapies.

Recombinant adeno-associated virus (AAV) has been widely used in preclinical and clinical studies to deliver therapeutic payloads to multiple tissues following systemic administration. A few naturally occurring AAV serotypes 8/9/rh10 are currently being employed for gene transfer to cardiac, musculoskeletal or central nervous system (CNS) tissues in clinical trials through different routes of administration (e.g., intravenous, intracoronary or directly into cerebrospinal fluid (intraCSF)) for Duchenne Muscular Dystrophy, X-linked myotubular myopathy, Pompe disease and Batten disease amongst others[1,2]. A commonly utilized benchmark in this regard is the AAV9 serotype, which is utilized in Zolgensma®, the first FDA approved

gene therapy to treat SMA by delivering a functional copy of the SMN1 gene following intravenous infusion in infants[3].

Similar to Zolgensma®, most AAV based gene therapy trials focused on correcting functional deficits in the CNS or skeletal muscle require high vector doses to achieve therapeutic efficacy[2]; however, this practice has recently been shown to carry the potential risk of unintended side effects[1,4]. For instance, in humans, AAV vectors administered systemically can result in transient elevation in liver transaminase levels in patient serum, complement activation, thrombotic microangiopathy, renal failure or in severe cases, death[5,6]. These aspects are only partially modeled (or sometimes not at all) or

[1]Department of Molecular Genetics and Microbiology, Duke University School of Medicine, Durham, NC, USA. [2]Department of Surgery, Duke University School of Medicine, Durham, NC, USA. [3]North Carolina State University College of Veterinary Medicine, Raleigh, NC, USA. [4]Department of Pediatrics, Duke University School of Medicine, Durham, NC, USA. [5]StrideBio Inc., Research Triangle Park, Durham, NC, USA. [6]Duke Regeneration Center, Duke University School of Medicine, Durham, NC, USA. [7]Department of Biomedical Engineering, Duke University, Durham, NC, USA. [8]These authors contributed equally: Katherine E. Simon, Leo O. Blondel. ✉e-mail: aravind.asokan@duke.edu

predicted by preclinical animal models. For instance, studies focused on dorsal root ganglion pathology following AAV vector treatments in NHP and pig models appear to directly correlate with AAV transduction efficiency and dose[7,8]; however, such toxicologic aspects have generally not been observed in mouse models, or reported in clinical trials such as in case of SMA to date. Similarly, no vector-triggered toxicity was observed in dose escalation studies utilizing AAVrh74 or AAV8 vectors over-expressing micro/minidystrophin in the mdx mouse or canine models, respectively[9,10]. Interestingly, other side effects such as renal toxicity and thrombocytopenia have been reported in clinical trials utilizing AAV9[2,4], but similar adverse events have not been reported to date with rh74 vectors[11]. While the exact mechanisms related to these toxicities remain under investigation, these recent findings highlight significant disparities in AAV biology across species, but also the need for improved vector design that can (i) mitigate potential risks and improve the therapeutic window associated with AAV gene therapies as well as (ii) enable better predictive modeling leading to effective clinical translation.

To this end, several AAV library-based approaches have been developed in the past decade. For instance, directed evolution has been utilized to engineer novel AAV capsids with improved CNS or muscle tropism, improved tropism for human tissue or the ability to evade neutralizing antibodies[12,13]. These efforts have yielded novel muscle or neurotropic AAV capsids such as the rationally engineered variants AAV2i8, AAV2i8G9[14,15];chimeric AAVB1, AAV1RX obtained through DNA shuffling of natural capsid isolates[16,17] or AAV-PHP.B, AAV.CAP.B10, AAVMYO, MyoAAV, amongst others obtained through capsid surface peptide display[18–21]. A notable drawback that has been reported is that directed evolution of AAV libraries in a single animal model can yield variants specific for those species, as exemplified in case of AAV-PHP.B[22], which utilizes the C57/B6 mouse strain-specific receptor, Ly6A[23]. Indeed, it remains to be seen whether evolving AAV libraries solely in NHPs yields only primate sub-species-specific variants and ultimately, whether NHP-derived AAV capsids are the most suitable candidates for clinical evaluation. To address these challenges, we describe a novel cross-species-based approach that can be readily implemented into any directed evolution workstream independent of the engineering strategy or library-based approach. Specifically, we iteratively cycled AAV libraries administered intravenously and isolates amplified from CNS tissue in pigs, mice, and non-human primates to generate cross-species compatible AAVs (ccAAVs). Gene transfer efficiencies of different enriched capsid variants were determined in mice, which yielded a lead capsid variant, AAV.cc47. We benchmarked the transduction efficiencies in mouse liver, CNS, cardiac and skeletal muscle tissues in a mouse model using a Cre recombinase cassette packaged within AAV.cc47 or AAV9 and administered at three different doses as well as increased acid alpha-glucosidase (GAA) expression in the mouse CNS, heart, and skeletal muscle with AAV.cc47 compared to AAV9. Additionally, we demonstrate higher genome editing efficiencies in the mouse heart and skeletal muscle using optimized CRIPSR/Cas9 cassettes in a reporter mouse model as well as the mdx mouse model for DMD. Further, we demonstrate the cross-species compatibility of AAV.cc47 compared to AAV9 in pigs, and nonhuman primates following intrathecal/cisterna magna dosing. Our approach corroborates the notion that iterative evolution in multiple animal models can yield potent AAV variants with cross-species compatibility that can potentially enable better predictive preclinical modeling and help improve the therapeutic window of AAV gene therapies in humans.

## Results

### Cross-species cycling of AAV capsid libraries

To engineer cross-species compatible AAVs (ccAAVs), we created AAV9-based capsid libraries and cycled them through pigs, mice, and non-human primates following systemic administration (Fig. 1a).

Libraries were created through saturation mutagenesis of the AAV9 capsid surface loop corresponding to amino acids (AA) 452–458 (VP1 subunit numbering). This region corresponds to variable region IV (VR-IV; red) of the capsid, which is the farthest radially protruding surface loop[24], and plays a multifunctional role in AAV biology, including a neutralizing antibody binding site[25,26] as well as transduction efficiency and cell surface receptor binding, due to proximity to the 3-fold-symmetry axis of the AAV9 capsid[24,25]. Briefly, binding interactions of galactosylated glycans and AAVR with the AAV9 capsid are known to occur around the 3-fold axis. In addition to the functional attributes outlined above, previous studies by our laboratory[27,28] and the Gradinaru laboratory[19] have demonstrated the importance of VR-IV in evolving novel AAV capsids with improved functionality. Moreover, the VR-IV loop is less permissive towards peptide insert approaches compared to VR-VIII. Taken together, VR-IV is an important surface region for incorporation into AAV capsid libraries. This region is highlighted in 3D structures of the VP3 subunit monomer, trimer and full capsid conformations to highlight the protruding nature of this surface loop (Fig. 1b). Wild-type AAV libraries packaging mutant genomes consisting of AAV2 *Rep* and AAV9 mutant *Cap*, flanked by AAV2 ITRs were first cycled in 3-week old piglets dosed at 1e13 vg/kg via intravenous (IV) administration. Pig brains were harvested 3 days post injection and further dissected into parietal, occipital, and temporal cortices, cerebellum, hippocampus, thalamus, hypothalamus, and medulla regions. Viral genomes amplified from extracted DNA were pooled and cloned to generate the next AAV library, which was administered IV in 8-week-old C57/B6 mice at a dose of 2e13 vg/kg. The brains from these mice were harvested 3-days post injection and dissected into cortex, dentate gyrus, hippocampus, and cerebellum regions; then processed and prepared as above. The final species used for in vivo cycling were 2-year-old cynomolgus macaques. Libraries were administered via intravenous infusion at a dose of 3.3e13 vg/kg. Macaque brains were harvested 7 days post injection and dissected into temporal, parietal, occipital, and frontal lobes, thalamus, corpus callosum, hippocampus, cerebellum, brainstem, pons, and midbrain regions. Amplified AAV genomes from macaques as well as from each species in earlier rounds were then prepared for next-generation sequencing.

Parental and evolved viral libraries were sequenced with an Illumina NovaSeq platform. Sequencing reads were analyzed using a custom in-house Perl script and plotted in R (Fig. 1c). Each bubble corresponds to an individual library amino acid sequence and bubble diameter corresponds to fold enrichment between parental and evolved libraries. Enrichment analysis of individual AAV clones following evolution in each animal species revealed a 100 to 69,000-fold enrichment between evolved variants and parental representation. The dominant lead variant AAV.cc47, accounted for 67% of the overall mapped reads after the final round. AAV.cc47 percent representation and fold enrichment and total library diversity was determined following evolution in each species (Supplementary Fig. 1). Using monkey next-generation sequencing (NGS) data, we performed a consensus motif analysis of the top 100 AAV clones from the final round. This analysis revealed 452-GVSLGGG-458 as the major consensus sequence of the top clones, which was determined to be the amino acid sequence for AAV.cc47 (Fig. 1d). We further analyzed the NGS data of parental and evolved (monkey) libraries by calculating the amino acid frequency at each position in our 7-mer libraries. Additionally, we determined the differential representation of amino acids at each library position between the two data sets. A clear selection for our consensus motif is corroborated by our evolved variant heatmap (Fig. 1e). The parental library shows a broad distribution of amino acids across positions 1–7, with a bias for glycine at each position illustrated by the yellow gradient. The least and most frequent amino acids are shown in blue and yellow, respectively, while the 50th percentile is shown in white. In order to assess the manufacturability of AAV.cc47

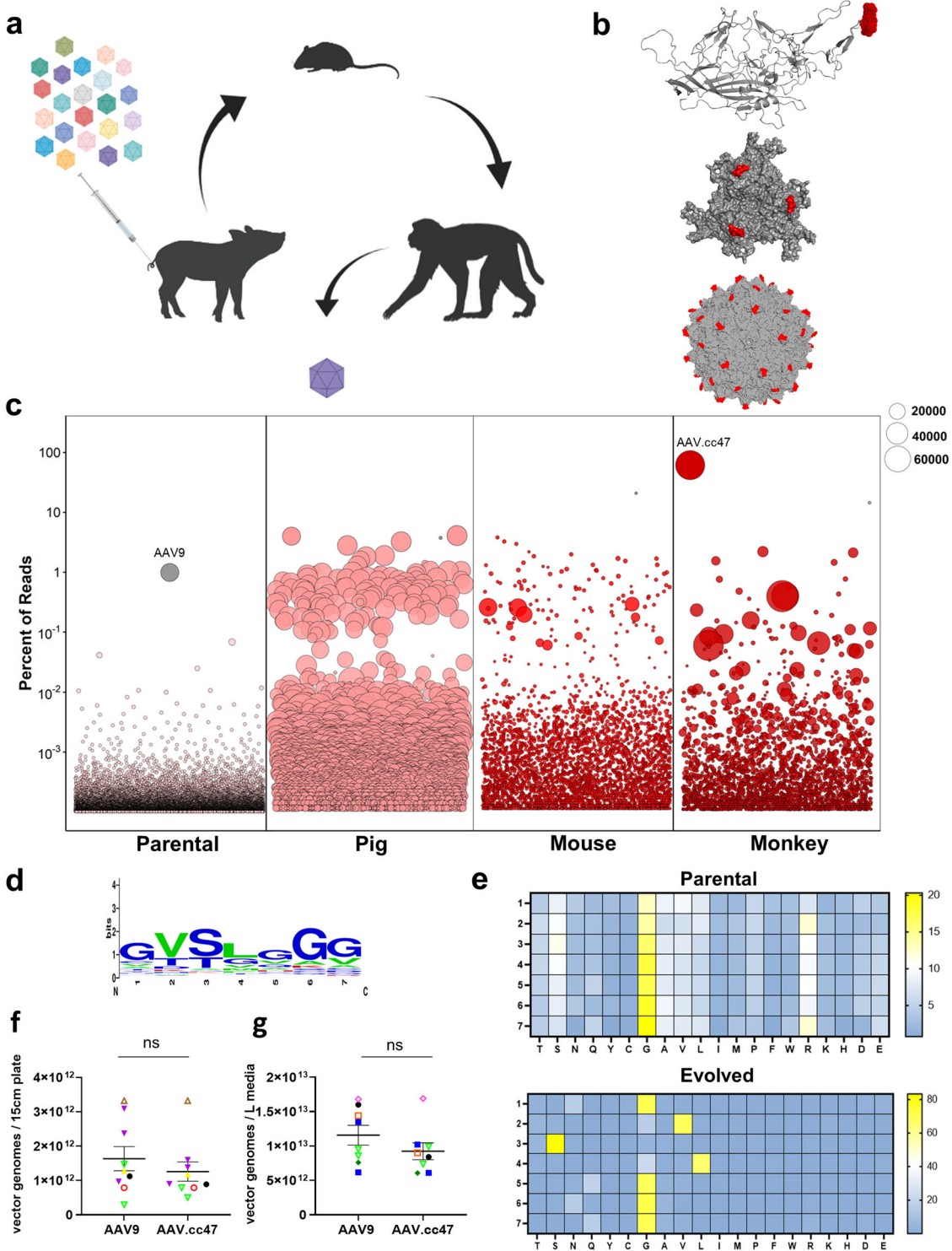

compared to AAV9, we collated titers obtained by qPCR analysis of yields from production runs of these two capsids packaging the same transgene cassettes produced in both adherent and suspension HEK293 systems. We found no significant difference in manufacturability of AAV.cc47 compared to AAV9 in either adherent or suspension production systems (Fig. 1f–g).

### AAV.cc47 is more potent than AAV9 vectors in normal mice

We first compared the transduction profiles of AAV.cc47 and AAV9 vectors packaging a self-complementary CBh-mCherry cassette following systemic administration at 5e13vg/kg at 4 weeks post-dosing

(Fig. 2a). Native mCherry fluorescence from tissue sections of injected mice show increased transduction efficiency in heart, tibialis anterior, and brain, while a modest, yet statistically significant increase in liver transduction was observed (Fig. 2b, c). Quantification of fluorescence intensity in the heart, tibialis anterior, and liver shows AAV.cc47 has a 21-, 16-, and 2-fold higher fluorescence intensity, respectively (Fig. 2d). Further, we observed increased transduction and spread of AAV.cc47 throughout the mouse brain compared to AAV9 (Fig. 2c). A greater number of neurons transduced in the cortex (CTX), cerebellum (CB), and hippocampus (HC) by the evolved variant was determined with 4-, 3-, and 4- fold greater transduction in this cell type, respectively,

**Fig. 1 | Cross-species evolution of AAV capsid libraries yields a dominant new variant. a** Schematic of AAV capsid library evolution in pigs, mice, and non-human primates following intravenous dosing **b** AAV9 VP3 monomer, trimer, and full capsid identifying the 7 amino acid residues mutated in our capsid library (red). The images were generated using PyMOL. **c** Next-generation sequencing identifies AAV.cc47 as the top enriched AAV variant following three rounds of evolution in each animal species. Each amino acid sequence was assigned a random number 1–1000 and is plotted on the x-axis, and the y-axis represents the percent of reads of each sequence in the library. The bubbles represent an individual amino acid sequence and its size corresponds to the fold enrichment of each sequence in the evolved library compared to the parental library. **d** Consensus motif analysis of the top 100 enriched AAV mutants following three rounds of evolution, for residues 452 – 458 (VP1 numbering). **e** Heat map of parental and final evolved libraries identifying amino acid frequencies at each library position following in vivo cycling of libraries. **f, g** Comparison of multiple recombinant AAV9 or AAV.cc47 production

yields in both adherent ($n = 9$) and suspension ($n = 15$) HEK293 systems. Final, post-purified AAV yields from adherent 293 s are plotted as vector genome titers per 15 cm plate and from suspension HEK293s are plotted as vector genome titers per liter of media. Data points represented here consist of vectors packaging the same transgene cassette for AAV9 and AAV.cc47. Each symbol and color combination represents a different transgene cassette used for vector preparation, the dash represents the mean value and error bars represent the standard error mean. Vectors that were single-stranded genomes include black circles, yellow triangles, open red circles, open brown triangles, and purple triangles. Vectors that were self-complementary genomes include open upside-down light green triangles, blue square, open orange square, dark green diamond, and open pink diamonds. Statistical significance was determined by a two-tailed paired Student $t$ Test (Adherent, $P < 0.081$; Suspension, $P < 0.056$). ns not significant. Source data are provided as a Source Data file.

compared to AAV9 (Fig. 2e). Representative co-staining images used for quantification can be found in Supplementary Fig. 2. The identities of other cell types transduced in the brain remain to be determined. Lastly, a 4-fold higher fluorescence intensity with AAV.cc47 compared to AAV9 for rostral, midbrain, and caudal regions was observed (Fig. 2d). Next, we compared AAV.cc47 and AAV9 for Cre recombinase expression in the Ai9 tdTomato fluorescent reporter mouse model. Briefly, AAV.cc47 and AAV9 vectors packaging a single-stranded CMV-Cre cassette were administered at 4e11, 4e12, or 4e13vg/kg IV in adult homozygous Ai9 reporter mice and tissues were harvested at 4 weeks post-injection (Fig. 3a). Immunofluorescence imaging corroborated earlier reporter gene expression studies with AAV.cc47 transducing Ai9 mouse heart and tibialis anterior 3- to 4-fold more effectively than AAV9, while showing no difference in liver transduction (Fig. 3b–d, Supplementary Fig. 3). A dose-dependent increase in the number of transduced cardiomyocytes and myofibers is observed between AAV.cc47 and AAV9, increasing by approximately an order of magnitude as we increase in dose. No such response was observed in hepatocytes, indicating vector dosage saturation in the liver (Supplementary Fig. 3). Surprisingly, vector genome biodistribution for all tissues analyzed showed no difference in biodistribution between AAV.cc47 and AAV9 at any of the three doses evaluated (Fig. 3e), while a dose dependent increase in vector genome copy numbers was observed in each tissue. Vector genome biodistribution was quantified in other tissues at each dose and revealed a similar profile (Supplementary Fig. 4). Enhanced transduction in the brain from our mCherry study was confirmed with our evolved variant at the 4e13 vg/kg dose of Cre Recombinase in Ai9 mice (Supplementary Figure 5a). In addition, increased targeting of neuronal cells in the brain with AAV.cc47 was also confirmed by immunofluorescence co-staining (Supplementary Fig. 5b, c).

### AAV.cc47 vectors demonstrate improved genome editing efficiency

We then wanted to explore whether IV dosing of AAV.cc47 packaging CRISPR/Cas9 and guide RNAs can improve genome editing efficiency compared to AAV9 in Ai9 mice (Fig. 4a). To determine optimal conditions for genome editing, we evaluated a panel of vector design strategies based on a dual AAV vector approach, which requires *Staphylococcus aureus* (Sa) Cas9 and two single guide RNAs (gRNAs) targeting the *Rosa26* locus in Ai9 mice. Similar to the Cre recombinase approach described earlier, this CRISPR-based approach is designed to remove the lox-STOP-lox region, which will then enable tdTomato expression in different tissues. In Design 1, we used single-stranded AAV genomes with the CMV enhancer/beta-actin (CB) promoter driving SaCas9 and a different gRNA driven by the hU6 promoter; mixed in a 1:1 ratio (dose = 1e14 vg/kg each vector). For Design 2, we utilized single-stranded AAV genomes with the cytomegalovirus (CMV) promoter driving SaCas9 in one vector and both gRNAs in the second

vector driven by the hU6 promoter; which were mixed in a 1:1 ratio (dose = 1.5e14 vg/kg each vector). Design 3 is similar to Design 2, except the vectors were mixed in a 1:3 Cas9-gRNA vector ratio (dose = 2e14 vg/kg each vector; Supplementary Fig. 6a–c); Design 4 is similar to Design 2, except that the guide RNAs are expressed within a self-complementary AAV genome and the Cas9 and guide vectors were mixed in a 1:1 ratio (dose = 2e14 vg/kg each vector; Fig. 4a). Following systemic administration, AAV.cc47 CRISPR/Cas9 dual vectors edited the Rosa26 locus at a higher efficiency than AAV9 in all 4 AAV-CRISPR designs evaluated. Notably, the self-complementary genome to deliver two single guide RNAs yielded the highest percentage of editing in the heart with a 3-fold increase in genome editing of the *Rosa26* locus with AAV.cc47 compared to AAV9, determined by the number of tdTomato positive cardiomyocytes normalized to DAPI (Fig. 4b). We also observed a 3-fold increase in genome editing in tibialis anterior with AAV.cc47 compared to AAV9 as shown by the increased number of tdTomato expressing myofibers (Fig. 4c). Consistent with reporter gene and Cre expression studies, we observed no significant fold change in genome editing efficiencies in hepatocytes with AAV.cc47 and AAV9 packaging any of the vector designs in the liver (Fig. 4d). Vector genome biodistribution corroborated past analysis in Cre studies, showing no difference in heart, tibialis anterior, and liver tissues between AAV.cc47 and AAV9 (Supplementary Fig. 6d). Interestingly, gRNA expression was 3- to 4-fold greater in all three tissues of AAV.cc47 treated mice compared to AAV9 (Fig. 4e). Based on these observations, it is tempting to speculate that AAV.cc47 may not have a binding or uptake advantage in tissues, but may benefit from a post-entry event compared to AAV9. SaCas9 expression profile was different and relatively low, and in some cases below the limit of detection, compared to gRNA expression (Fig. 4f).

### AAV.cc47 transduces NHP brain and heart at higher efficiency than AAV9

To assess cross-species compatibility, AAV9 or AAV.cc47 vectors packaging a scCBh-mCherry cassette were dosed at 1e13vg via intra-cisterna magna (ICM) infusions in two-year-old cynomolgus macaques and harvested tissue 13 days post-injection (Fig. 5a). Immunohistochemical staining against mCherry in NHP brain revealed robust expression and spread in the monkey cortex and cerebellum with AAV.cc47 compared to AAV9, while penetration into the brain parenchyma was also greater in AAV.cc47 treated monkeys (Fig. 5b). Neuronal and glial transduction in the cortex was also higher with our evolved variant compared to AAV9 and a greater number of mCherry positive Purkinje neurons were observed in AAV.cc47 treated cerebellums (Fig. 5b). Expression was observed in peripheral tissues outside the access of the CSF, such as the heart, where AAV.cc47 demonstrated enhanced transduction compared to AAV9, and the liver (Fig. 5c, d). Biochemical analysis of tissues further corroborates the enhanced transduction phenotype seen in histology of NHP tissues

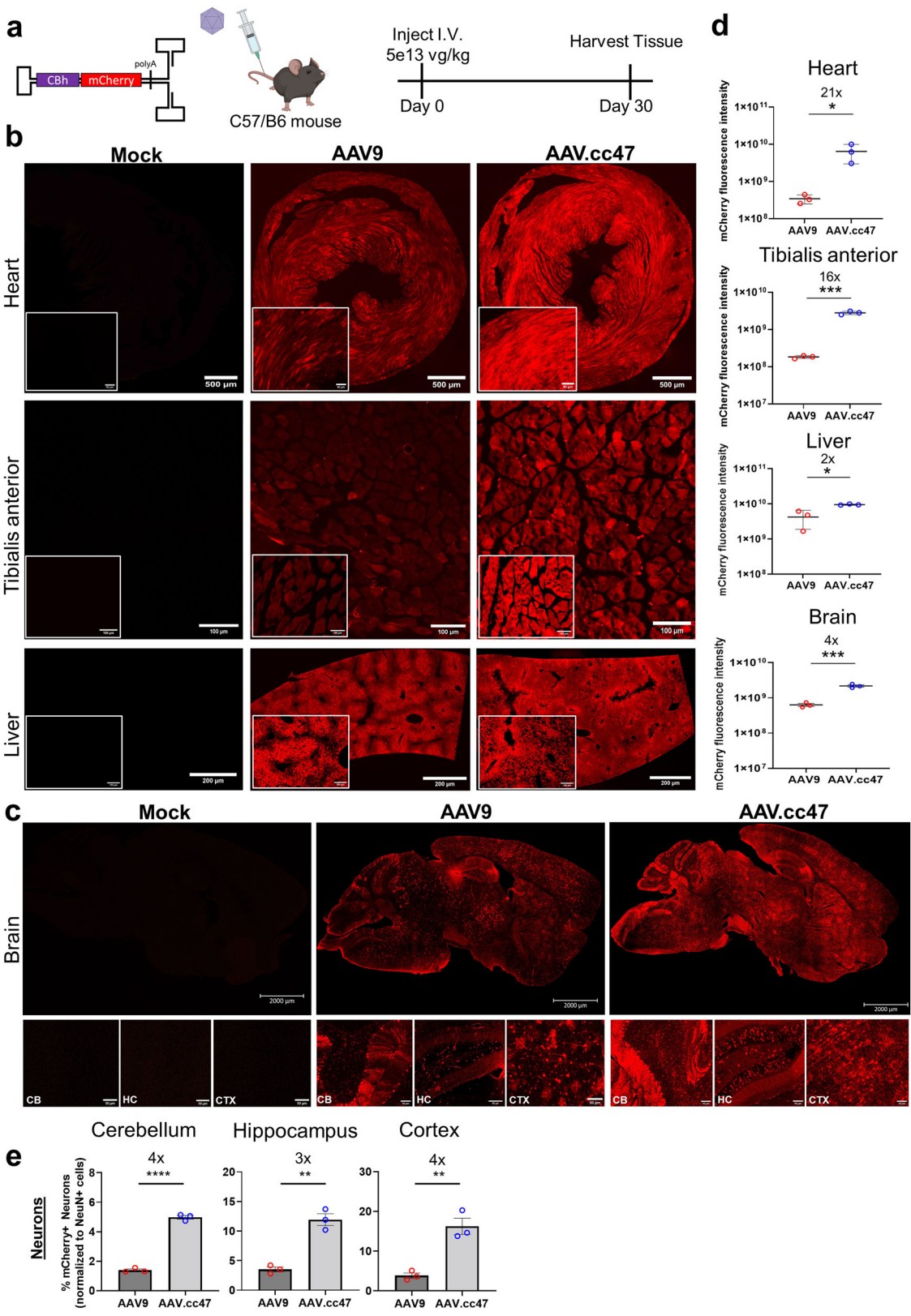

with AAV.cc47 (Supplementary Fig. 7a–f). This was most notable in the premotor cortex, where mCherry protein levels quantified by ELISA show a 10-fold high protein expression in this brain region with AAV.cc47 compared to AAV9. Equal vector genome biodistribution between both capsids was further corroborated in NHPs following ICM dosing (Supplementary Fig. 7a, d). Images of tissue from both AAV-injected NHPs have been provided (Supplementary Figs. 8–10).

Analysis of the lumbar and thoracic spinal cord regions by immunohistochemical staining of mCherry reveals greater transduction of cell types in these two spinal cord regions with AAV.cc47, while DRG expression was observed for both AAV9 and AAV.cc47 (Supplementary Fig. 10).

We wanted to evaluate the transduction of AAV.cc47 in other species as well following intraCSF dosing. Briefly, P0-1 C57/B6

**Fig. 2 | AAV.cc47 transduces mouse heart, skeletal muscle, and brain more efficiently than AAV9 following systemic administration. a** 8–10 week old C57/B6 mice (*n* = 3) were injected intravenously at a dose of 5e13vg/kg with AAV9 (red) or AAV.cc47 (blue) and organs were harvested 4 weeks post injection. Vectors delivered a self-complementary AAV genome with the chicken-beta actin hybrid (CBh) promoter driving mCherry reporter expression. **b** Representative images of native mCherry fluorescence in heart, tibialis anterior and liver. **c** Representative images of native mCherry fluorescence in whole brain, cerebellum (CB), hippocampus (HC), and cortex (CTX). **d** Quantification of native mCherry fluorescence intensity in heart (*P* < 0.0386), tibialis anterior (*P* < 0.0001), liver (*P* < 0.0172), and brain (*P* < 0.0172). Brain mCherry fluorescence intensity was measured from rostral, midbrain, and caudal brain regions, and are summarized as a single brain tissue. **e** Quantification of percent neurons transduced in the CB (*P* < 0.0001), HC

(*P* < 0.0015), and CTX (*P* < 0.0045) of C57/B6 mice was determined via immunofluorescence staining of mCherry and NeuN proteins and taking the ratio of total mCherry+ neuron counts to total NeuN+ cell counts in each brain region. Fifty micrometers thick cross sections were obtained via a vibratome for the heart, liver, and brain, while 7 μm thick cross sections were obtained for tibialis anterior via cryostat. For all quantification, 2 sections per mouse and 3 images per section (a total of 6 images) were used to quantify fluorescence intensity and cell counts using ImageJ. Each dot represents an individual mouse for fluorescence intensity, and for cell counting each dot represents an individual mouse, fold change is listed above significance, the dash and bars represent the mean value and error bars represent the standard error mean. Statistical significance was determined by a two-tailed Student *t* Test. **P* < 0.05; ** *P* < 0.01; ***P* < 0.001; ****P* < 0.0001; ns not significant. Source data are provided as a Source Data file.

neonates were injected via intracerebroventricular (ICV) administration of 2e10vg of AAV.cc47 or AAV9. Vectors packaged a self-complementary genome containing CBh-mCherry, and mice were sacrificed 4 weeks post injection (Supplementary Fig. 11a). Immunofluorescence staining of brain cross-sections was performed with an anti-rabbit mCherry primary antibody and show increased transduction and spread throughout the mouse brain of AAV.cc47 treated mice compared to AAV9, especially in the cortex, hippocampus, and cerebellum, (Supplementary Fig. 11b).

We also evaluated transduction in the pigs following intraCSF dosing of AAV.cc47 or AAV9 vectors packaging a self-complementary genome containing CBh-eGFP. Briefly, we administered 3.3e13vg via intrathecal (IT) infusion into the lumbar cistern of 3-week-old piglets and harvested tissue 13 days post injection (Supplementary Fig. 11c). Pig brains were immunohistochemically stained with an anti-rabbit eGFP primary antibody and corroborate the findings from our NHP studies of increased transduction and spread throughout the cortex and cerebellum of AAV.cc47 treated mice (Supplementary Fig. 11d). A markedly greater number of neuronal and glial cells transduced in these brain regions with AAV.cc47 compared to AAV9 was observed. These results collectively illustrate that AAV.cc47 is a cross-species compatible vector capable of greater transduction in the brain of mice, pig, and NHPs.

### AAV.cc47 demonstrates increased GAA transgene expression in a mouse model of Pompe disease

Following the evaluation of AAV.cc47 in wild-type mice and demonstrated cross-species compatibility, we investigated the ability of AAV.cc47 vectors to deliver therapeutic cargo in a mouse model of disease. To this end, AAV.cc47 and AAV9 vectors were compared in the acid alpha-glucosidase knockout (*Gaa*−/−) mouse model of Pompe disease. Mice were dosed IV (1.3e14vg/kg) with AAV vectors packaging a single-stranded genome encoding a functional copy of *GAA* driven by the CBh promoter (Fig. 6a). Analysis of GAA enzyme levels in brain tissues at 4 weeks post-dosing not only revealed that AAV.cc47-GAA is superior to AAV9-GAA vectors, but also enabled enzyme activity levels to reach 67% of WT mice compared to only 16% of WT levels in AAV9-GAA treated mice (Fig. 6b). In different segments of the spinal cord, GAA activity from AAV.cc47-GAA treated mice were nearly identical to that of WT mice, and activity from AAV9-GAA treated mice was only 26% of WT (Supplementary Figure 12a). In mouse heart and tibialis anterior, only AAV.cc47-GAA achieved significant increases in enzyme activity over all three other groups evaluated, *Gaa* (−/−) mock, AAV9-GAA, and WT (Fig. 6b). Consistent with earlier results, enzyme activity assays in the liver were nearly equal between AAV.cc47-GAA and AAV9-GAA vectors with both expressing supraphysiological levels (Fig. 6b). No statistical difference was observed in vector genome copy numbers (biodistribution) between AAV.cc47 and AAV9 in any of these tissues (Fig. 6c, Supplementary Fig. 12b).

### AAV.cc47 mediated gene editing demonstrates improved dystrophin restoration in the mdx mouse model

Having determined an optimal dual AAV-CRISPR cassette design strategy consisting of a single-stranded vector expressing CMV-SaCas9 and a self-complementary gRNA vector, we further exploited this approach to edit the mouse *Dmd* locus. Briefly, we replaced the gRNAs targeting the *Rosa26* locus with gRNAs targeting introns 22 and 23 of *Dmd*[29]. CRISPR/gRNA cassettes packaged in AAV.cc47 or AAV9 were administered IV at 1.4e14 vg/kg each vector (1:1 ratio) in Pax7-nGFP mdx mice (Fig. 6d). These mice harbor a premature stop codon in exon 23 which prevents normal expression of dystrophin and upon excision of exon 23, full-length dystrophin expression is restored. In the mdx mouse heart, AAV.cc47-CRISPR treatment results in 6-fold higher dystrophin expression, in relation to laminin expression, compared to AAV9 (Fig. 6e, g). In the tibialis anterior of mdx mice, AAV.cc47-CRISPR treatments results in a 2-fold higher restoration of dystrophin than AAV9-treated animals (Fig. 6f, g). Genome editing outcomes from both vectors were determined via Taqman quantitative RT-PCR by quantifying the percent of exon 23 deleted *Dmd* transcripts in each tissue. AAV.cc47 mediated genome editing resulted in 3- and 5-fold more exon 23 deleted transcripts in heart and tibialis anterior, respectively, compared to AAV9 (Fig. 6h). Quantification of vector genomes and gRNA expression corroborate our previous findings that there is no measurable difference in vector genome biodistribution to tissues, while increased gRNA expression is observed in AAV.cc47 treated tissue (Supplementary Fig. 13).

### Discussion

Multiple CNS, cardiac and skeletal muscle-focused gene therapies being evaluated in the clinic utilize AAV9 vectors[2,30]. The widespread adoption of this vector to deliver therapeutic transgenes is due in part to the broad tissue tropism and abundance of transduction data in different preclinical models. Previous efforts by our lab have demonstrated the plasticity of capsid surface spike regions, particularly VR-IV and VR-VIII, through structure-guided evolution of different AAV serotypes to achieve improved transduction, tropism, and antibody evasion[27,28]. In this study, we markedly improved the transduction profile of AAV9 by evolving capsid libraries generated by saturation mutagenesis of VR-IV residues in three different animal models (pigs, mice, and non-human primates). As alluded to earlier, this multispecies approach was designed to specifically address the concern that newly evolved/engineered AAV variants may not readily translate from mouse models of disease to large animal models for toxicity/biodistribution studies and ultimately to humans in a scalable and predictable fashion.

At the structural level, we observed a striking shift toward amino acids with hydrophobic or no side chains in different positions N452G, G453V, S454S, G455L, Q456G, N457G, and Q458G. The newly evolved 452-GVSLGGG-458 did not impact capsid stability based on production yields with a range of cassettes. Ongoing efforts to further understand

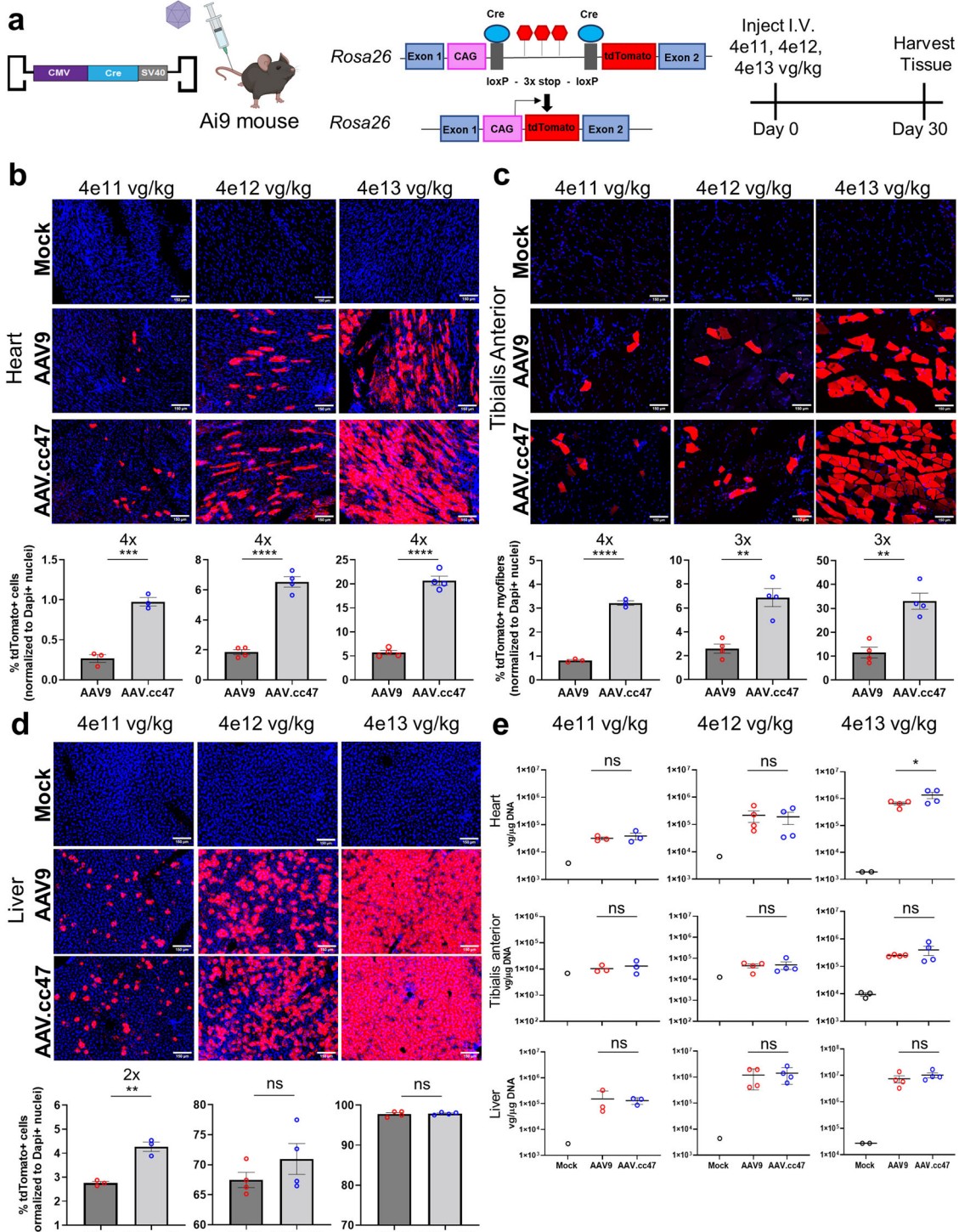

**Fig. 3 | AAV.cc47 outperforms AAV9 in Cre recombination in the Ai9 reporter mice at all tested doses following intravenous (IV) administration. a** 8–10 week old Ai9 reporter mice ($n = 4$) were injected intravenously at 4e11, 4e12, and 4e13 vg/kg with AAV9 (red) or AAV.cc47 (blue). Vectors delivered a single stranded AAV genome with the cytomegalovirus (CMV) promoter driving Cre recombinase. Representative immunofluorescence images and quantification for tdTomato (red) and DAPI (blue) for mice injected at each dose in heart (**b**), tibialis anterior (**c**), and liver (**d**). Fifty micrometers thick cross sections were obtained via a vibratome for heart and liver, while 7 μm thick cross sections were obtained for skeletal muscle via cryostat. Total number of tdTomato+ cells (heart and liver) and total number of tdTomato+ myofibers (tibialis anterior) were counted and normalized to the total number of DAPI + nuclei in each tissue. **e** Vector genome copy numbers per μg DNA were calculated by normalizing Cre recombinase copy numbers to total μg DNA

input for qPCR quantification and are represented as log vg/μg DNA. Each dot represents an individual mouse, fold change is listed above significance, the dash and bars represent the mean value and error bars represent the standard error mean. Quantifications in panels B-E are listed in ascending order of dosing from left to right, starting with 4e11 vg/kg and going to 4e13 vg/kg for each tissue type. Statistical significance was determined by One-Way ANOVA with Tukey's post-test (3 treatment groups) for vector genome biodistribution analysis (High dose heart, $P < 0.0279$). An unpaired two-tailed Student $t$ Test (2 treatment groups) was used for histological quantification (High dose heart, $P < 0.001$; tibialis anterior, $P < 0.0018$; liver, $P < 0.7883$; Mid dose heart, $P < 0.0001$; tibialis anterior, $P < 0.0023$; liver, $P < 0.2666$; low dose heart, $P < 0.0006$; tibialis anterior, $P < 0.0001$; liver, $P < 0.0019$). *$P < 0.05$; **$P < 0.01$; ***$P < 0.001$; ****$P < 0.0001$; ns not significant. Source data are provided as a Source Data file.

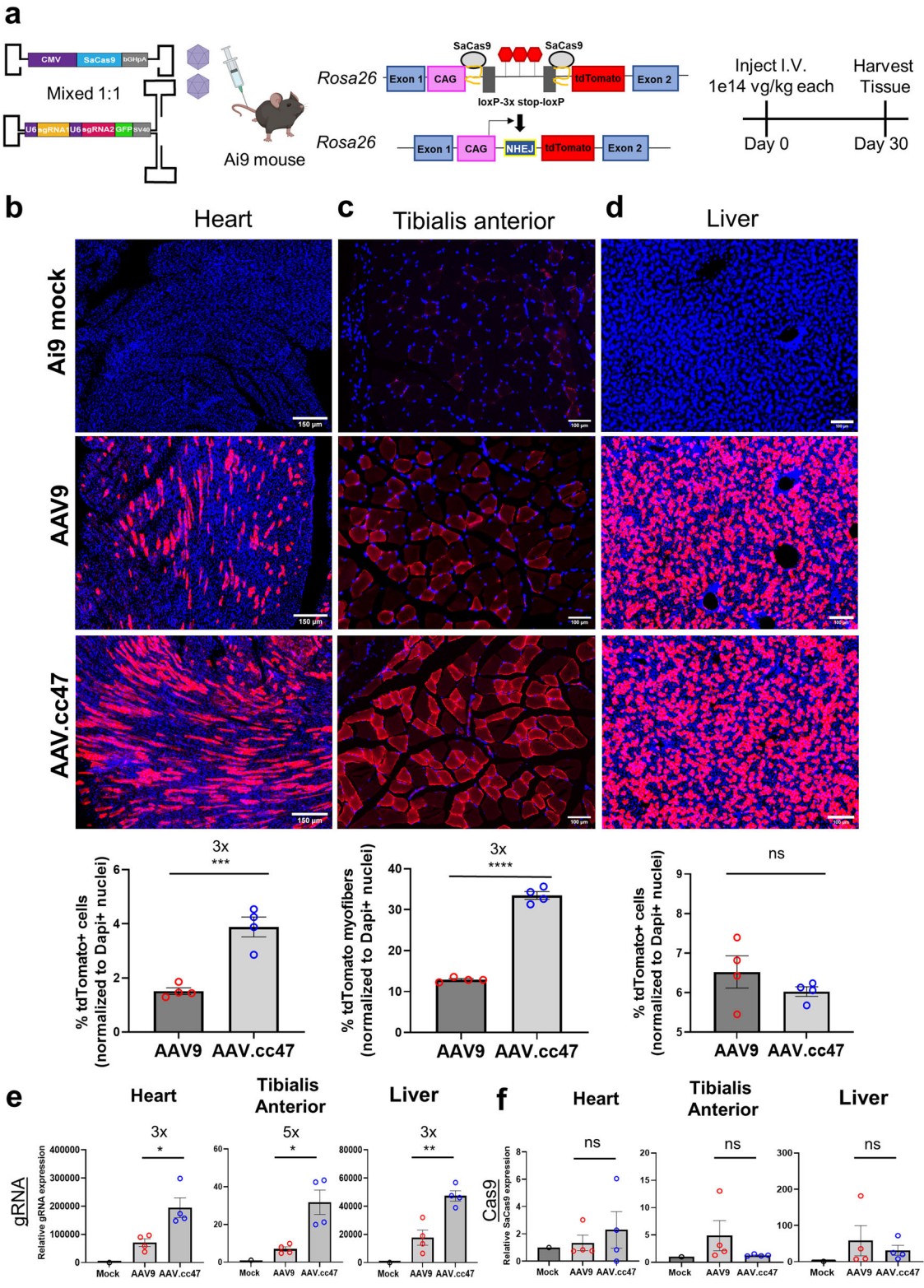

other AAV.cc47 attributes also include assessment of antigenicity as determined by neutralizing antibody assays. It is important to note that the AAV9 VR-IV surface loop has been previously implicated as an immunodominant epitope in antibody recognition[25,26]. Assessment of the immunogenicity of AAV.cc47 compared to AAV9 will require a large NHP cohort in a separately designed study. We have, however, evaluated the ability of AAV.cc47 to evade neutralizing antibodies (NAbs) from pig, NHP and human antisera (Supplementary Figure 14a–c). Our studies indicate that AAV.cc47 is neutralized similar

to AAV9 and corroborates earlier observations[27] that the newly evolved antigenic epitope in VR-IV alone is not sufficient for NAb evasion. However, it is important to note that, unlike the parental AAV9 surface epitope VR-IV, this newly evolved epitope is devoid of charged or aromatic surface exposed side chains, thereby presenting minimal features generally thought to be essential for receptor or immune interactions.

The cross-species evolution approach revealed several interesting trends. At a macro level, AAV library cycling leads to decreased

**Fig. 4 | Systemic administration of AAV.cc47-CRISPR increases genome editing efficiencies in heart and skeletal muscle of Ai9 reporter mice compared to AAV9. a** 8–10 week old Ai9 reporter mice (*n* = 4) were injected with AAV.cc47 (blue) or AAV9 (red) using our lead dual AAV vector system consisting of a single-stranded genome with cytomegalovirus (CMV) driven SaCas9 and a self-complementary genome with 2 gRNAs driven by U6 promoters targeting the *Rosa26* locus. The total dose administered was 2.0e14 vg/kg (1.0e14 vg/kg each vector). Representative immunofluorescence images for tdTomato (red) and histological quantification in heart (**b**), skeletal muscle (**c**), and liver (**d**). Fifty micrometers thick cross sections were obtained via a vibratome for the heart and liver, while 7 μm thick cross sections were obtained for skeletal muscle via cryostat. Quantification of the percentage of tdTomato+ cells (heart and liver) and myofibers (tibialis anterior) was normalized to the number of DAPI+ nuclei in each tissue. For all histological

quantification, 2 sections per mouse and 3 images per section (total of 6 images) were used to quantify % transduced cells and myofibers using ImageJ. Quantitative RT-PCR was performed with primers amplifying each gRNA (**e**) or SaCas9 (**f**) mRNA relative to mouse *Actb*. Each dot represents an individual mouse, fold change is listed above significance, the bar graphs represent the mean value and error bars represent the standard error mean. Statistical significance was determined by a two-tailed Student *t* Test (2 treatment groups; heart, *P* < 0.0009; tibialis anterior, *P* < 0.0227; liver, *P* < 0.2892) or One-Way ANOVA with Tukey's posttest (3 treatment groups; heart gRNA, *P* < 0.0314, Cas9, *P* < 0.7966; tibialis anterior gRNA, *P* < 0.0348, Cas9, *P* < 0.4459; liver gRNA, *P* < 0.0126, Cas9, *P* < 0.8205). *\*P* < 0.05; \*\**P* < 0.01; \*\*\**P* < 0.001; \*\*\*\**P* < 0.0001; ns not significant. Source data are provided as a Source Data file.

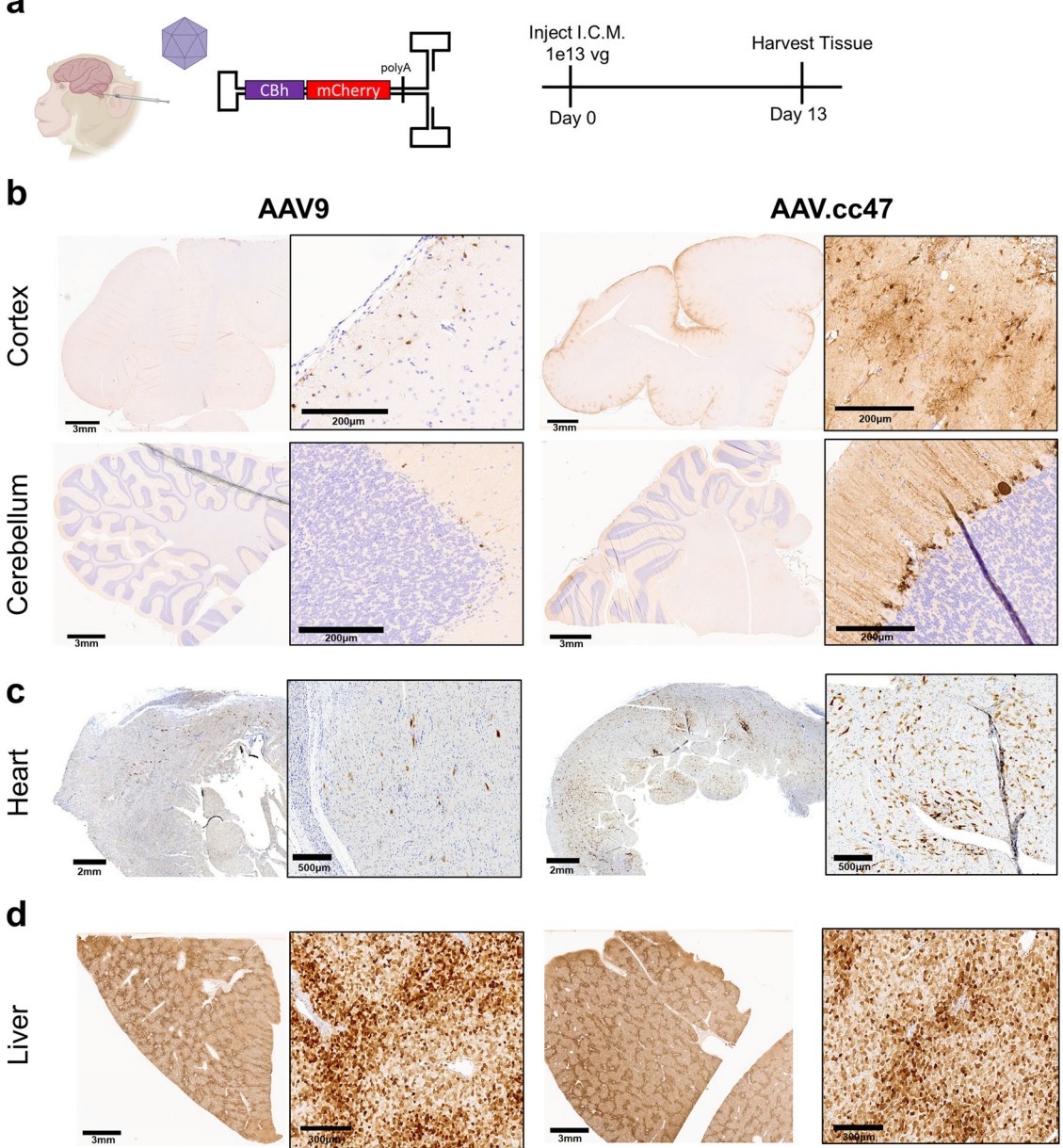

**Fig. 5 | AAV.cc47 transduces non-human primate (NHP) brain and heart more effectively than AAV9 following intracisternal magna (ICM) infusion. a** ICM infusion of 1e13 total vg of AAV9 or AAV.cc47 vectors in 2-year-old cynomolgus macaques (*n* = 2). Organs were harvested at 13 days post infusion. Vectors packaging self-complementary AAV genome with a truncated chicken beta-actin (CBh)

promoter driving mCherry expression were utilized for the study. Representative images of immunohistochemically stained NHP brain (**b**), heart (**c**), and liver (**d**) are shown. Tissue was embedded in paraffin following fixation and cut 5 μm thick and 3,3′-Diaminobenzidine (DAB) was used to visualize mCherry protein in tissues. Source data are provided as a Source Data file.

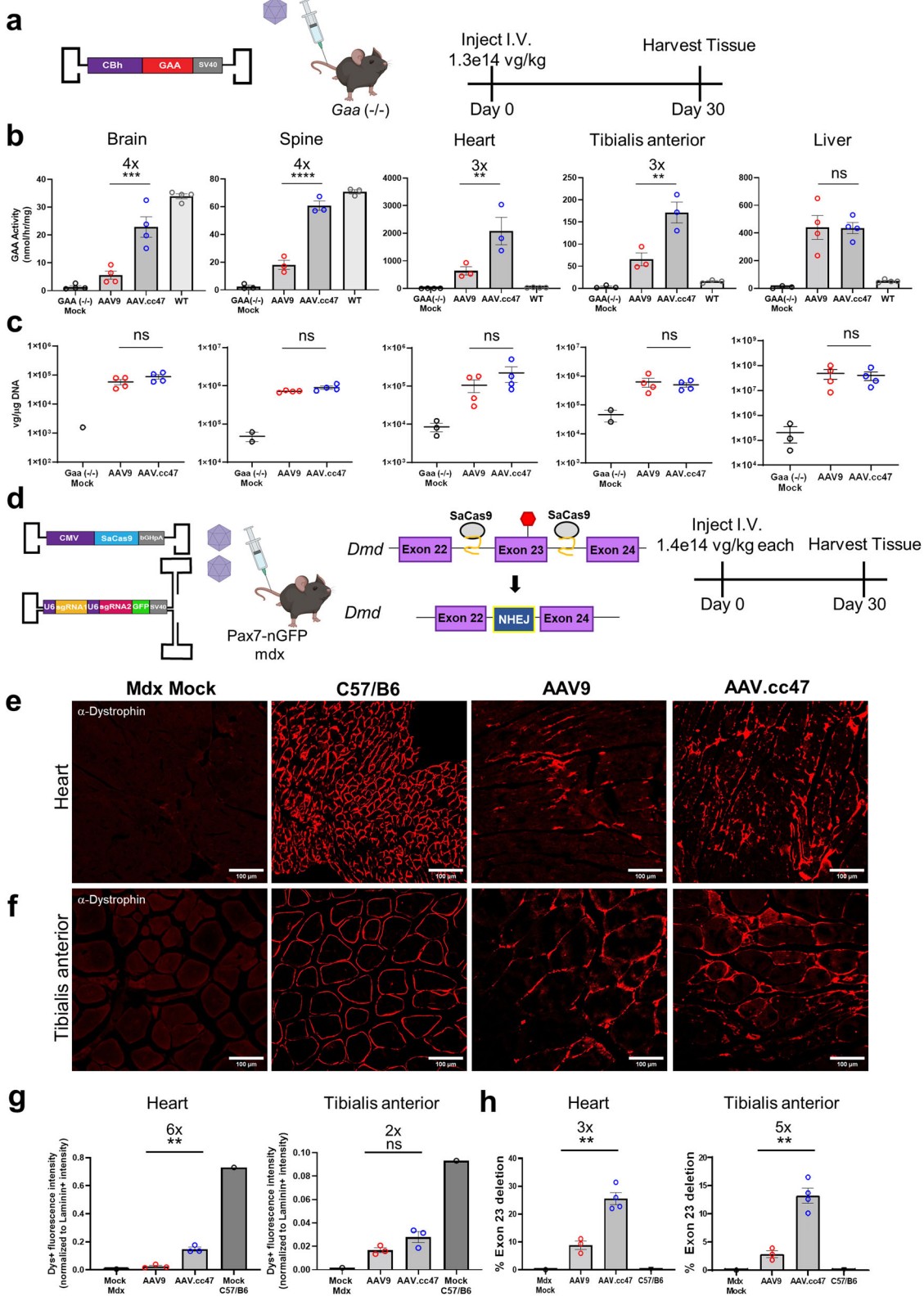

sequence diversity in each cycling step. Whether this phenomenon arises from the number of cycles alone or is also influenced by a specific animal model is unclear. Nonetheless, it is interesting to note that the percent representation of conserved amino acid residues for each position in VR-IV increases, particularly after cycling in NHPs. In particular, we observed significant enrichment of neutral amino acid residues with no to small side chains (esp. glycine, alanine, valine) in VR-IV as we cycled across each species. The final NHP cycling step

markedly enriched Gly, Ser and Leu residues within VR-IV. Amongst these, the Gly residues were preferably enriched in multiple species. Novel receptor or host factor usage and other functional ramifications of these structural changes are as yet unknown. It is, however pertinent to mention that AAV.cc47 is still AAVR-dependent for cellular infection (similar to AAV9; Supplementary Fig. 1d).

High-throughput sequencing data revealed that the AAV.cc47 variant started to emerge as a promising candidate after evolving the

**Fig. 6 | Systemic administration of AAV.cc47 leads to increased Acid alpha-glucosidase (*GAA*) expression in the Pompe mouse model and enhanced restoration of dystrophin through genome editing in mdx mice compared to AAV9. a** Pompe study schematic. 8–10 week old *Gaa* (−/−) mice (n = 4) were injected with 1.3e14vg/kg of a single stranded AAV genome expressing the human acid alpha-glucosidase gene driven by the CBh promoter. **b** GAA enzyme activity was quantified in tissues of AAV9 (red) or AAV.cc47 (blue) treated *Gaa* (−/−) mice. Data is represented as nmol per hour activity per mg BSA protein. **c** Vector genome copy numbers per μg DNA were calculated by normalizing SV40 polyA copy numbers to total μg DNA input for qPCR quantification and are represented as log vg/μg DNA. GAA enzyme levels for different brain and spinal cord regions are summarized in analysis. **d** Mdx study schematic. 8–10 week old Pax7-nGFP mdx mice (*n* = 4) were injected with a total dose of 2.8e14 vg/kg consisting of a single-stranded genome expressing SaCas9 driven by the CMV promoter and a self-complementary genome expressing two gRNAs driven by the U6 promoter mixed in a 1:1 ratio. Representative immunofluorescence images for dystrophin (red) in the heart (**e**) and tibialis anterior (**f**) from AAV treated Pax7-nGFP mdx mice. Ten micrometres thick and 7μm thick cross sections were obtained via a cryostat for the heart and tibialis anterior, respectively. **g** Quantification of dystrophin expression intensity normalized to laminin expression intensity in heart and tibialis anterior. **h** Quantification of exon 23-deleted transcripts in cardiac and skeletal muscles via Taqman quantitative RT-PCR. Statistical significance was determined by One-Way ANOVA with Tukey's posttest (GAA enzyme brain, $P < 0.0003$; spine, $P < 0.0001$, heart, $P < 0.0062$, tibialis anterior, $P < 0.0029$, liver, $P < 0.9999$; Dystrophin histology quantification heart, $P < 0.0057$, tibialis anterior, $P < 0.2700$; % *Dmd* exon 23 deletion heart, $P < 0.0075$, tibialis anterior, $P < 0.0056$). Data points included represent an individual mouse for all assays. Fold changes are shown above significance and all graphs represent the mean value and error bars represent the standard error mean. *$P < 0.05$; **$P < 0.01$; ***$P < 0.001$; ****$P < 0.0001$; ns not significant. Source data are provided as a Source Data file.

AAV library in pigs, stringent selective pressure on our AAV capsid libraries was also observed in the non-human primate brain. This was evidenced not only by percent representation but also by the fold enrichment of AAV.cc47 compared to the parental library following evolution in this species. We acknowledge that the current study only demonstrates improved potency of AAV.cc47 in CNS, cardiac and liver tissue in NHPs, when administered through the cisterna magna. These observations were recapitulated in the pig brain, where due to their size and amount of vector required, the animals were dosed intrathecally as well. Thus, a critical aspect of AAV.cc47 biology that we continue to explore is the ability to transduce multiple tissues at high efficiency as well as toxicity studies in large animal models following different routes of administration (IV, intracoronary etc.).

AAV.cc47 demonstrated superior gene transfer efficiency in multiple organs compared to AAV9 in three different mouse models as evidenced by reporter gene expression, Cre recombinase and GAA enzyme activity. Striking differences were observed in the brain, heart, and tibialis anterior, while the liver appeared to demonstrate saturated transgene expression at the doses tested, corroborating broad tissue tropism similar to AAV9. These results corroborate that AAV.cc47 can potentially improve the therapeutic window for certain gene therapy applications. Surprisingly, no difference in vector copy number or tissue biodistribution were observed between the two AAV vectors. Since our evolution approach did not select for capsids with improved tropism, but rather enhanced transduction efficiency, the comparable biodistribution of AAV.cc47 and AAV9 in all tissues and in different species is thought-provoking. It is also noteworthy to mention that we observed improved transgene expression, but similar biodistribution regardless of the cassette (ss vs sc) or promoter utilized. At this writing, it is tempting to speculate that AAV.cc47 has evolved to exploit a post-entry mechanism that may account for increased potency, while maintaining tissue uptake similar to AAV9. For instance, intracellular trafficking steps such as improved endosomal escape, nuclear entry, capsid uncoating/genome release and/or transcriptional efficiency could contribute to the observed capsid profile. Further studies are warranted to dissect a plausible mechanism of action.

AAV-mediated genome editing faces several significant challenges[31,32]. Of particular relevance, our understanding of the correlation between delivery, CRISPR/Cas9 and guide RNA (gRNA) expression and editing efficiency continues to evolve. In the current study, we developed an optimal configuration of dual AAV vector systems required, while concurrently evaluating the editing efficiencies afforded by AAV.cc47 and AAV9 vectors. In Ai9 mice, we observed that higher gRNA to Cas9 expressing vector ratios or the use of self-complementary AAV cassettes to express gRNAs in conjunction with an improved capsid (AAV.cc47) markedly improved gene editing efficiency. These results corroborate the notion that gRNA levels might be rate limiting in in vivo genome editing strategies[33,34]. Analysis of RNA transcript levels of both SaCas9 and gRNAs in these tissues revealed that AAV.cc47 vectors indeed transduced tissues more efficiently than AAV9 vectors further lending support to an improved post-entry mechanism. Of note, while the amount of SaCas9 mRNA transcripts observed at 4 weeks was low post-injection, gRNA levels were found to be relatively high. This disparity in RNA levels between SaCas9 and gRNA could arise from promoter strength differences (Pol-II vs Pol-III), CMV promoter silencing in muscle, an immune response to SaCas9 in adult mice, orAAV genome composition (single-stranded SaCas9 vs self-complementary gRNA cassettes). Further, previous studies have shown that the CMV promoter is prone to silencing in muscle[35] and this has been observed in AAV-CRISPR muscle genome editing as well[29]. It is plausible that an immune response to Cas9 in adult mice and promoter silencing could have contributed to this effect. The U6 promoter-driven gRNAs do not appear to be subject to this at least at earlier time points reported here. It should also be noted that overall editing efficiencies in adult cardiac and skeletal muscle tissues (but not the liver) following IV dosing of dual AAV vector systems show room for improvement in general. Nevertheless, our studies unequivocally demonstrate that AAV.cc47 significantly improves genome editing efficiency with CRISPR/Cas9 systems; however, we acknowledge that additional improvements to cassette design might further enhance desired outcomes.

In summary, our study corroborates that cross-species evolution can yield highly potent AAV variants with the ability to transduce multiple tissues across different preclinical animal models. Notably, we test different mouse strains, cynomolgus macaques and/or pigs, all of which constitute critical path in the clinical development of different human gene therapies. The lead capsid, AAV.cc47 is currently being evaluated for a broad range of therapeutic gene transfer applications in preclinical models of musculoskeletal, cardiac, CNS and kidney disease. While mechanistic insights continue to be explored, we postulate that subjecting AAV capsid libraries to evolutionary pressure in different species is likely to comprehensively address both host-specific and host-agnostic factors (essential or restrictive) that impact AAV transduction. This cross-species approach can be readily adapted to any AAV library evolution/screening and, when combined with promoter-specific selection strategies, potentially improve the clinical translatability of next-generation AAV vectors.

## Methods
### Study design
The overall goal of this study was to determine whether cycling of AAV capsid libraries through multiple animal species would yield a cross-species compatible AAV variant with improved properties. The general approach was to select preclinically relevant animal models to cycle our AAV libraries through and directly compare gene transfer and genome editing efficiencies benchmarked against the parental AAV9 serotype. For all dosing strategies, injections were blinded for each test vector evaluated. In the case of the Pompe study, tissue

analysis from different cohorts was blinded. All mouse and pig protocols were approved by the Institutional Animal Care and Use Committee (IACUC) at Duke University (mouse; Protocol A189-21-09) and North Carolina State College of Veterinary Medicine (pigs; Protocol 20-425). NHP studies and protocols were reviewed and approved by IACUC at Southern Research (Birmingham, AL; Protocol 15863.01) and CR-MWN IACUC at Charles River Labs (Kalamazoo, MI; Protocol 2728-017), both accredited by the Association for Assessment and Accreditation of Laboratory Animal Care-International (AAALAC). All studies were carried out using applicable Standard Operating Procedures at Southern Research (Birmingham, AL) and Charles River Labs (Kalamazoo, MI) and previously reported by our lab[36]. During the study, the care and use of animals was conducted in accordance with the guidelines of the USA National Research Council, the US Department of Agriculture (Animal Welfare Act; Public Law 99–198) and those of the Guide for the Care and Use of Laboratory Animals (National Academies Press, 2011). In general, sample size for mouse studies ranges from 3–6 replicates for each condition per experiment, with specific information provided in the figure legends and materials and methods section. In the case of our large animal studies, 1-2 replicates were used for each species.

### AAV plasmid libraries

The AAV9 VR-IV plasmid library was constructed as previously described[27,28]. Briefly, the parental library plasmid pTR-wtAAV9-VR-IV-library consisting of AAV2 ITRs flanking AAV2 Rep and AAV9 *Cap* genes was generated by randomizing a portion of *Cap* corresponding to VR-IV (amino acids 452–458 in *Cap*) via saturation mutagenesis. The subsequent evolved library plasmids differed from the parental library plasmid only in the generation of the library inserts. Prior to the incorporation of the randomized insert containing degenerate nucleotides at the selected region into the parental library, a region of *Cap* encompassing VR-IV was replaced with a partial GFP sequence to prevent any incorporation of WT VR-IV sequences in the parental library. The parental library insert was incorporated into the WT AAV plasmid containing a region of GFP via overlap extension of 2 PCR products. PCR and overlap extension reactions were performed using Q5 polymerase (New England Biolabs). The plasmid backbone and library insert were then restriction enzyme digested with XbaI and BsiWI and ligated together. Purified ligation products were electroporated into DH10B ElectroMax cells (Thermo Fisher Scientific) and directly plated on multiple 245 × 245 mm2 bioassay dishes to avoid bias from bacterial suspension cultures. Plasmid DNA from pITR-AAV9 libraries were purified from pooled colonies grown on LB agar plates.

### AAV capsid library and recombinant vector production

AAV capsid libraries were generated as previously described[27,28]. In brief, adherent HEK293 cells (obtained from the University of North Carolina Vector Core) were transfected in 15 cm dishes (Corning) with polyethylenimine (PEI; Sigma-aldrich) in a 1:3 DNA:PEI ratio. Equal molar ratios of pITR-AAV9 Library plasmid and adenovirus helper plasmid pXX680 were used during transfection. Cell media was harvested and replaced 4 days post transfection, followed by the final media harvest on the day 6. Recombinant AAV vectors produced in adherent HEK293 cells were performed as previously described[37]. Recombinant AAV vectors produced in suspension HEK293 cells (generated in-house at Duke University) were generated via a triple plasmid transfection method consisting of an adenovirus helper plasmid pXX680 (0.60 µg/mL), AAV9 or AAV.cc47 *Rep/Cap* plasmid (0.50 µg/mL), and ITR plasmid (0.30 µg/mL). Plasmids were transfected into cells with PEI (1 to 3 ratio, plasmid to PEI). Media was harvested 6 days post transfection and cells were pelleted via centrifugation and discarded. Cell media was incubated with 12% Polyethylene glycol (PEG) for 48 h at 4 C. Precipitated capsids were then collected via centrifugation and resuspended in formulation buffer

containing 1 mM MgCl2 and 0.001% F-68. Resuspended capsids were DNase treated for 1 hr at 37 C prior to iodixanol gradient purification via an ultracentrifuge. Next, an iodixanol (Sigma-Aldrich) gradient consisting of 60%, 40%, 25%, and 17% densities were created manually starting from the bottom in this order. DNased PEG suspension in that added to the top of the gradient and loaded in the ultracentrifuge. Gradients were allowed to spin at 30,000 RPM and 17 C for at least 6 h. Upon completion, gradients were fractioned into 550 µL aliquots before quantification of AAV genomes and downstream purification. AAV concentrations in each iodixanol fraction were determined via qPCR using primers that bind against a portion of the ITR region (Supplementary Table 1). Iodixanol fractions with the most concentrated titers were selected for subsequent desalting and buffer exchange using a Zeba spin desalting column (Thermo Fisher Scientific), according to manufactures instructions. Final titer amounts of each AAV library prep was also determined via qPCR with primers against the ITRs (Supplementary Table 1). For CSF dosing via intrathecal and intracisternal magna infusions, recombinant AAV vectors were further processed to remove existing endotoxins using a Pierce high-capacity endotoxin removal spin column (Thermo Fisher Scientific). Endotoxin levels were then quantified via a Chromogenic LAL endotoxin assay kit (GenScript).

### Cross-species cycling of AAV capsid libraries

AAV capsid libraries were produced as described in the section above. The first round of AAV evolution occurred in the pig brain following intravenous administration. The brain was harvested and dissected further into multiple brain regions prior to genomic DNA isolation. AAV genomes were amplified from pig genomic DNA using primers targeting AAV9 *Cap* (Supplementary Table 1). PCR amplicons containing library sequences were ligated into our AAV library plasmid backbone and used to generate the round 2 AAV9 library. The second round of AAV evolution occurred in mice following intravenous administration. Mouse brains were harvested 3 days post injection and dissected further into multiple brain regions. AAV9 *Cap* containing the library region was amplified through PCR and cloned into our AAV library plasmid backbone prior to production of the next round AAV libraries. The final round of AAV evolution in non-human primates occurred following intravenous administration of the last AAV library preparation. Brains from these primates were harvested 7 days post injection and were processed as described for pigs and mice. In vivo cycling between these three species was completed before next-generation sequencing of capsid variants.

### High-throughput sequencing analysis and identification of newly evolved AAV strains

Sequence diversity of the parental and evolved AAV9 capsid libraries were assessed by using the viral library preparations. The evolved library sequences were generated following PCR of DNA amplified tissue followed by plasmid cloning and viral production. Both the parental and evolved viral libraries after production were DNase-I treated prior to extraction of viral genomes from the capsids and subsequent addition of Illumina adapter sequences via PCR. The first round of PCRs adds half of the Illumina sequencing adapters using primers specific to the amplicon and the second round of PCR adds the remaining adapter sequences and index sequence. After each round of PCR, the products were purified using the PureLink PCR Micro Kit (Invitrogen). The quality of the amplicons was verified using a Bioanalyzer (Agilent), and concentrations were quantified using a Qubit spectrometer (Thermo Fisher Scientific). Libraries were then prepared for sequencing with the Illumina NovaSeq 6000 S-Prime Reagent Kit (300 cycles), following manufacturer's instructions, and sequenced on the Illumina NovaSeq system. Demultiplexed reads were analyzed with an updated in-house Perl script[28]. Reads are surveyed for nucleotide sequences flanking the library regions and intermediate sequences are

counted and ranked. These nucleotide sequences are then translated, and the resulting amino acid sequences are also counted and ranked. Then, the percent representation and fold enrichment between parental and evolved libraries were calculated and ranked. Amino acid sequences were plotted according to their percent representation and enrichment for each library. A second Perl script was used to calculate the frequency of each amino acid at each position in the library, considering each capsid variant[28]. Perl scripts were executed using Strawberry Perl for Windows software v 1.32.1.1. Bubble plots representing capsid mutants and amino acid distribution heatmaps were generated using the R studio graphics package v3.5.2.

### AAV packaging plasmid constructs
The plasmids scAAV-CBh-mCherry and scAAV-CBh-eGFP were generated or obtained as described previously[27,28,36]. The plasmid ssAAV-CMV-Cre-bG used for our dose escalation studies in Ai9 mice was generously provided by Dr. Charles Gersbach (Duke University). For AAV-CRISPR studies, constructs containing the CMV enhancer/beta-actin (CB) promoter driving Staphylococcus aureus (Sa) Cas9 and gRNAs targeting the mouse *Rosa26* locus were provided by Dr. William Lagor (Baylor College of Medicine; Supplementary Table 1) through the Somatic Cell Genome Editing Consortium (SCGE). The same gRNA 1 and 2 were both used in all Ai9 CRISPR studies. For AAV-CRISPR studies where both gRNAs are on a single vector, both gRNAs and U6 promoters used in the CB strategy were cloned into a single-stranded AAV plasmid containing 2 gRNA scaffolds[29] via Gibson assembly (NEB). For Ai9 CRISPR studies using a self-complementary gRNA vector, the dual gRNA insert from the ssAAV gRNA plasmid was cloned into the plasmid scAAV-CBh-mCherry replacing the CBh-mCherry insert with the gRNA insert via Gibson assembly. For mdx CRISPR studies, gRNA sequences targeting the intronic regions around exon 23 in *Dmd* were used as previously reported[29]. See Supplementary Table 1 for the *Dmd* targeting gRNA 1 and 2 sequences. The ssAAV-CMV-SaCas9 plasmid used in Ai9 and mdx studies was generously provided by Dr. Charles Gersbach (Duke University). The ssAAV-CBh-GAA plasmid used in *Gaa* (−/−) mice were generously provided by Dr. Mai K Elmallah (Duke University).

### Mouse studies
All mouse strains used in this study were maintained at Duke University School of Medicine with the assistance of Duke's Division of Laboratory Animal Resources (DLAR). Mice were housed in a temperature-controlled (-18–23 C, 40–60% humidity) and enriched environment, with a 12 h light/dark cycle, and provided standard chow and water.

### Intravenous dosing in wild-type and reporter mice
All mouse studies utilized male and female adult mice aged 8–10 weeks old, and AAV was administered systemically via the tail vein for all experiments. The wild-type C57/B6 mouse colony was bred and maintained at Duke University. Ai9 reporter mice on a C57/B6 background were originally purchased from Jackson Laboratories (Jax#007909)[38], and a breeding colony was established at Duke University. AAV evolution in C57/B6 mice were injected systemically with 2e13vg/kg of AAV. Brains were harvested 3 days post injection prior to library amplification. For mCherry reporter studies (*n* = 3), C57/B6 mice were injected systemically with 5e13vg/kg of AAV. Our dose escalation studies (*n* = 3) were done in Ai9 reporter mice injected with 4e11, 4e12, or 4e13 vg/kg of AAV vectors packaging CMV driven Cre recombinase. Gene editing studies targeting the *Rosa26* locus were performed in Ai9 mice. For CB driven SaCas9 studies (*n* = 4), 5e13vg/kg each vector was administered. For CMV driven SaCas9 studies with a 1 to 1 ratio (*n* = 3), 7.5e13 vg/kg each vector was administered. For a 1 to 3 cas9 to gRNA ratio (*n* = 5), 5e13 and 1e14 vg/kg, respectively, were administered. For CMV driven SaCas9 studies with a self-complementary gRNA vector (*n* = 4), 1e14 vg/kg each vector was administered systemically. Brian, spinal cord, heart, tibialis anterior, and liver were harvested 4 weeks post injection for all gene transfer and editing studies in C57/B6 and Ai9 mice.

### Intravenous dosing in disease mouse models
All genome editing mouse studies utilized only male mice, while gene transfer studies in *Gaa* (−/−) utilized male and female adult mice. All mice were aged 8–10 weeks at time of injection and AAV was administered systemically via the tail vein for all experiments. Pax7-nGFP mdx mice[39] on a C57/B6/SJL background were generously provided by Dr. Charles Gersbach (Duke University) and *Gaa* (−/−) mice[40] on a B6:129 background were generously provided by Dr. Mai K Elmallah (Duke University). For AAV-CRISPR studies in Pax7-nGFP mdx mice (*n* = 3), 1.4e14 vg/kg of each vector (Cas9 and gRNA) was administered intravenously into male mice. Gene transfer studies in *Gaa* (−/−) were evaluated at 1.3e14 vg/kg with AAV.cc47 or AAV9. Brian, spinal cord, heart, tibialis anterior, and liver were harvested 4 weeks post-injection for all gene transfer and editing studies in mdx and *Gaa* (−/−) mice.

### Intracerebroventricular (ICV) dosing in wild-type neonatal mice
All mouse ICV dosing occurred in in-house bred P0-P1 male and female C57/B6 neonates (*n* = 6). Pups were rapidly anesthetized on ice for 2 min followed by ICV injection using a stereotaxic apparatus. AAV vectors (2 μL total volume) packaging the CBh-mCherry transgene cassette were injected into the left lateral ventricle using a Micro-injection Syringe Pump (World Precision Instruments) at a controlled rate of 0.066 μL per second with a Hamilton 700 series syringe and 26 s gauge needle attached to a KOPF-900 small animal stereotaxic instrument (KOPF instruments). The following stereotaxic coordinates were used for injections: 0.8 mm relative to the sagittal sinus, 1.5 mm rostral to transverse sinus, and 1.5 mm deep. Dosing in these mice consisted of a total of 2e10vg administered in 2 μL. Mice were revived using a heating pad and rubbed in the bedding after AAV injections before being placed back with the dam. Mouse brains were harvested 4 weeks post infusion.

### Pig evolution and dosing studies
Pigs used in this study were a cross between Landrace, Yorkshire, and Duroc strains. Pig studies consisting of AAV evolution or reporter expression were all performed in 3-week-old newly weaned piglets weighing approximately 7 kgs. For AAV evolution in pigs (*n* = 2), male and female piglets were injected systemically via the heart vena cava with 1e13 vg/kg of AAV libraries. Brain and spinal cord were harvested 3 days post injection prior to AAV library amplification. For intrathecal reporter studies in pig, 3-week-old male pigs (*n* = 1) weighing approximately 7–10 kgs were anesthetized via a ketamine:xylazine cocktail containing 20 mg/kg ketamine +2 mg/kg xylazine. AAV vectors (2 mL total volume) packaging a CBh-eGFP transgene cassette were injected into the lumbar cistern with a total of 3e13vg and were infused at a rate of 1 mL per minute. Pigs brains were harvested 13 days post infusion.

### Non-human primate evolution and dosing studies
Female Cynomolgus macaques aged 2–3 years old were used for our AAV evolution studies. These primates were injected intravenously with 3.3e13vg/kg of AAV libraries. Primates were sacrificed 7 days post injection and target organs were harvested prior to library amplification. For reporter studies in non-human primates (*n* = 2), female 2–3 year-old Cynomolgus macaques were administered a total of 1e13vg in a total volume of 2 mLs via intracisternal magna infusion. Target organs from injected primates were harvested 13 days post infusion.

### Histological processing and staining
All C57/B6 and Ai9 tissues isolated were post fixed in 10% formalin (VWR) overnight. Livers, hearts, and brains were washed 3x in 1xPBS,

embedded in 3% agarose and sectioned 50 µm thick via a vibratome (Leica Biosystems). Tibialis anterior skeletal muscles were incubated in 30% sucrose overnight following fixation, embedded in O.C.T compound (Electron Microscopy Sciences), and frozen in liquid nitrogen-cooled isopentane. O.C.T. blocks were sectioned 7 µm thick on a Leica cryostat. For mCherry evaluation, tissue was mounted on slides with ProLong Gold Antifade Mountant with DAPI (Thermo Fisher Scientific) and imaged for native fluorescence. To quantify the percent of trans-duced neurons in the brain, tissue was incubated in blocking solution (5% normal goat serum, 0.1% Triton X-100 in 1x PBS) for 1 h at room temperature followed by immunofluorescence staining of brain tissue with anti-chicken mCherry (1:750; Abcam) and anti-rabbit NeuN (1:500, Abcam) primary antibodies overnight at 4 C. Tissue was then washed 3x in 1xPBS and incubated with anti-chicken Alexafluor 488 (1:500; Invitrogen) and anti-rabbit Alexafluor 647 (1:500; Invitrogen) second-ary antibodies for 1 h. For immunofluorescence staining in Ai9-Cre and Ai9-CRISPR studies, tibialis anterior skeletal muscles were incubated in citrate buffer (Abcam) at 80 C for 25 min and allowed to cool to room temperature in cool water. Heart, tibialis anterior, liver, and brain sections were then incubated in blocking solution, containing 5% normal goat serum and 0.1% Triton X-100 in 1xPBS, for 1 h at room temperature. Brains were further incubated in Mouse on Mouse (M.O.M.) blocking reagent (Vector labs) for 2 h. Heart, tibialis anterior, liver, and brain sections were then incubated with an anti-rabbit RFP primary antibody (1:500; Rockland Immunochemicals), and in addi-tion, brains were also incubated with anti-mouse NeuN primary anti-body (1:500; Abcam), diluted in blocking buffer overnight. Tissue sections were washed 3x in 1xPBS the following day and incubated with anti-rabbit Alexafluor 647 (1:500; Invitrogen) and anti-mouse Alexa-fluor 488 (brain only; 1:500; Invitrogen) secondary antibodies. For Ai9-Cre studies, tissue was then washed 3x in 1xPBS followed by secondary antibody incubation with an anti-rabbit Alexafluor 488 (1:1000; Invi-trogen) and anti-mouse Alexafluor 647 (brain; 1:500; Abcam) second-ary antibodies in addition to DAPI (Thermo Fischer Scientific). For Ai9-CRISPR studies, tissue was washed 3x in 1xPBS followed by secondary antibody incubation with an anti-rabbit Alexafluor 647 (1:1000; Abcam) and DAPI (Thermo Fischer Scientific). For immuno-fluorescence staining of ICV injected mouse brains, brain sections were prepared for staining as described above except sections were incu-bated with an anti-rabbit mCherry primary antibody (1:750; Abcam) and anti-rabbit Alexafluor 647 secondary antibody (1:500; Abcam). Immunostained tissue sections were then mounted on slides with ProLong Gold antifade reagent without DAPI (Invitrogen).

All mdx mouse tissues isolated were post fixed in 10% formalin (VWR) overnight. Mdx hearts and tibialis anterior skeletal muscles were then incubated in 30% sucrose overnight following fixation, embedded in O.C.T compound (Electron Microscopy Sciences), and frozen in liquid nitrogen-cooled isopentane. O.C.T. blocks were sec-tioned 10µm (heart) and 7µm (tibialis anterior) thick on a Leica cryo-stat. For immunofluorescence staining in mdx CRISPR studies, tibialis anterior skeletal muscles were incubated in citrate buffer (Abcam) at 80 C for 25 min and allowed to cool to room temperature in cool water. Heart and tibialis anterior sections were then incubated in a blocking solution, containing 5% normal goat serum and 0.1% Triton X-100 in 1xPBS, for 1 h at room temperature. Next, tissues were incubated in Mouse on Mouse (M.O.M.) blocking reagent (Vector labs) overnight. Tissue was then incubated with anti-mouse dystrophin (1:500; MANDSY8; Sigma-Aldrich) and anti-rabbit laminin (1:300; Abcam) primary antibodies diluted in a blocking buffer overnight. Next day, tissue was washed 3x with 1xPBS followed by secondary antibody incubation with an anti-rabbit Alexafluor 488 (1:500; Invitrogen) and anti-mouse Alexafluor 647 (1:500; Abcam), followed by DAPI staining (Thermo Fisher Scientific). Immunostained tissue sections were then mounted on slides with ProLong Gold antifade reagent without DAPI (Invitrogen).

All NHP and pig tissues isolated were post fixed in 10% formalin (VWR) overnight. Brain, heart, liver, spinal cords, and DRGs were embedded in paraffin following fixation and cut 5 µm thick. For immunohistochemical (IHC) staining of monkey brains, tissue sections were deparaffinized through a series of graded alcohols and xylenes before being loaded into Leica Bond RX system for staining. ER1 (citrate buffer was used for HIER) primary antibodies used were an anti-rabbit mCherry (NHP; 1:750; Abcam) and anti-rabbit eGFP (pig; 1:750; Abcam). Signal was detected using a Bond Refine Polymer detection kit (Leica biosystems) for a 1 h incubation followed by 5-min DAB (3, 3'-diaminobenzidine) to visualize bound mCherry and eGFP as a brown precipitate.

### Histological quantification and analysis
In order to quantify total fluorescence intensity for our mCherry and Cre reporter studies, the following equation was used: total fluores-cence intensity = Integrated Density – (Area of tissue region X mean fluorescence of background readings). To quantify the percent mCherry+ neurons in the brain, the total number of mCherry+ neurons were counted in the cerebellum, hippocampus, and cortex and counts were normalized to the total number of NeuN+ cells quantified manually in ImageJ v 1.52a. To quantify the percent tdTomato+ cells from our Ai9-Cre and Ai9-CRISPR studies, the total number of tdTo-mato+ cells in liver and heart and tdTomato+ myofibers in tibialis anterior were quantified manually and were normalized to the total number of DAPI + nuclei quantified automatedly in ImageJ. The per-cent tdTomato+ neurons in the brain from our Ai9-Cre studies was determined as described above for our mCherry study. The amount of dystrophin restoration was quantified by measuring the fluorescence intensity of dystrophin+ cardiac and skeletal muscles and normalizing to the fluorescence intensity of laminin+ tissue. The equation to determine total fluorescence intensity was used to measure the intensity from each image set as done for our mCherry and Cre studies. Then the ratio for dystrophin and laminin fluorescence intensity was determined for each tissue. All mouse staining quantification was performed using at least two sections and taking three images each section for all mice, with at least $n = 3$ for each study.

### Determination of vector genome biodistribution by quantita-tive PCR
DNA was extracted from tissue using a Purelink Genomic DNA Mini kit (Thermo Fisher Scientific) following manufactures instructions. Vector genomes were quantified via qPCR, using a plasmid standard and primers targeting AAV cassette elements (Supplementary Table 1). The biodistribution of viral genomes is represented as the ratio of vector genomes per microgram of DNA extracted.

### Determination of AAV vector transcripts by quantitative RT-PCR
RNA was extracted using TRIzol Reagent (Invitrogen) following the manufacturer's protocol. RNA was resuspended in nuclease-free water and stored at −80 °C until use. 5 µg of extracted RNA was subjected to DNase digestion using the TURBO DNA-free Kit (Ambion). Equal nanogram amounts of DNased RNA was used for cDNA synthesis using the High Capacity RNA-to-cDNA kit (Applied Biosystems). The pro-duced cDNA was used for quantitative PCR using Ai9 sequence-specific primers for gRNA 1 and gRNA2 or *Dmd* sequence-specific primers for gRNA1 and gRNA2, both with a common reverse primer specific to the gRNA scaffold, SaCas9, mCherry, and mouse *Actb* (Supplementary Table 1). Quantitative RT-PCR was carried out using a Roche Light-Cycler 480 and SYBR Green Mastermix (Roche Applied Sciences).

### Taqman-based determination of *Dmd* exon 23 deletion by quantitative RT-PCR
RNA extraction, DNase digestion, and cDNA synthesis were performed as described for quantification of AAV vector transcripts. The

produced cDNA was used with probes against the *Dmd* exon 4-5 junction to quantify total *Dmd* transcripts and another probe against *Dmd* exon 22–24 junction to quantify the amount of *Dmd* exon 23-deleted transcripts ([41]; Supplementary Table 1). A commercial Taqman assay for 18S ribosomal RNA was used as a housekeeping control (ThermoFisher). Each plate was run in duplicate with all three probes on a single plate for each tissue. Taqman Fast Advanced Master Mix (LifeTech) was used following manufacturers instructions for cycling conditions. Delta-Ct values between exon 4–5 and exon 22–24 were used to quantify the percentage of *Dmd* exon 23 deleted transcripts compared to total *Dmd* transcripts.

### Acid alpha-glucosidase (GAA) activity assay
GAA enzymatic activity was quantified as previously described[42,43]. Briefly, snap-frozen tissues were homogenized using a FastPrep bead system (MP Biomedicals), followed by three cycles of freezing and thawing. The samples were then clarified by centrifugation. Samples were evaluated in duplicate in a black 96-well plate. 20 μL of the sample was incubated with 4-methylumbelliferyl α-D-glycopyranoside substrate (Sigma-Aldrich) at 37 °C for 1 h in sodium acetate (Sigma-Aldrich) buffer at pH 4.3. The reaction is stopped with sodium carbonate (Sigma-Aldrich) buffer pH 10.7. Fluorescence was read with a Varioskan LUX (Thermo Fisher Scientific) at 360 nm excitation and 460 nm emission. The values were compared to those from a standard curve made with 4 mM methylumbelliferyl (Sigma-Aldrich). Samples were normalized to total protein content measured by DC Protein Assay (Bio-Rad), performed as described by the manufacturer.

### Determination of mCherry protein levels by ELISA
mCherry protein levels were quantified with a commercially available colorimetric ELISA kit (Abcam). Briefly, a tagged capture antibody and HRP-labeled detection antibody were mixed in solution with standards, controls, or samples generating a capture antibody/mCherry/detection antibody complex which is then immobilized to the wells of a 96-well plate coated with an anti-tag antibody. 3,3′,5,5′-Tetra-methylbenzidine (TMB) substrate was added, the reaction was stopped, and absorbance was measured at 450 nm.

### Luciferase expression assay
HuH7 scramble and AAVR knockout cell lines[44] were generated in-house at Duke University. Cells were seeded in 24-well plates at 50,000 cells per well in DMEM supplemented with 10% FBS and pen strep. The cells were then transduced with AAV9 or AAV.cc47 vectors packaging a single stranded CBA-luciferase reporter at a MOI of 10,000, 50,000, and 100,000 vector genomes per cell and incubated at 37 °C for 48 h. Cells were then lysed in 100 μL Passive Lysis Buffer (Promega), and transduction was assessed by measuring luciferase transgene expression with a Victor X3 microplate reader (PerkinElmer) and luciferase assay reagent (Promega) per manufacturer specifications.

### AAV Neutralizing antibody assay
Antiserum (from different species, diluted 1:5) was mixed with an equal volume of recombinant AAV9-Luc or AAV.cc47-Luc vectors and then incubated at room temperature for 30 min in tissue culture–treated, black, glass-bottom, 96-well plates (Corning). Then, HEK293s were seeded at 10,000 cells/well on top of the incubated vectors. Vectors were used at a MOI of 100,000 vg/cell and transduced cells were incubated in 5% CO2 at 37 °C for 24 h. Cells were then lysed with 25 μL of 1× passive lysis buffer (Promega) for 30 min at room temperature. Luciferase activity was measured on a VICTOR3 multilabel plate reader (PerkinElmer) immediately after the addition of 25 μL of luciferin (Promega). All readouts were normalized to controls with no antibody/antiserum treatment. Recombinant AAV vectors packaging CBA-Luc transgenes, antibodies, sera, were prediluted in DMEM and used in this assay.

### Statistical analysis
Data are represented as mean ± standard error mean. For data sets with two groups, significance was determined using an unpaired student's *t* test unless otherwise noted. For data sets with three groups, significance was determined by one-way ANOVA, with Tukey's post-test. Where indicated, ns represents not significant; $*p < 0.05$; $**p < 0.01$; $***p < 0.001$; $****p < 0.0001$; ns not significant.

### Reporting summary
Further information on research design is available in the Nature Research Reporting Summary linked to this article.

## Data availability
The NGS datasets for capsid libraries reported in this article are available under Sequence Read Archive accession code PRJNA869670. All other data associated with this study are present in the paper or the Supplementary Materials. Source data for each relevant figure is provided in a Source Data file. The data that support the findings of this study are available from the corresponding author upon reasonable request. Correspondence and requests for materials should be addressed to AA at aravind.asokan@duke.edu. Source data are provided with this paper.

## Code availability
The Perl scripts used for NGS analysis of AAV variants have been described previously[28]. The scripts have been deposited into Zenodo repository and can be accessed via https://doi.org/10.5281/zenodo.7075694. Correspondence should be addressed to AA at aravind.asokan@duke.edu.

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

## Acknowledgements

This study was funded by NIH grants awarded to AA (UG3AR07336, UH3AR075336) in addition to support from Translating Duke Health. We would like to acknowledge A. Rosales, D. Fiflis, S. Stuppy and M. L. Huston for their assistance with molecular biology, L. Edwards and K. Gleason for their assistance in completing in-life portions for our pig studies, N. Olby for her assistance in performing the intrathecal infusions in pigs, J. McNamara and E. Matthews for their assistance with dissection of pig brains, H. Daniels for her assistance with breeding the Pax7-nGFP-mdx mice used in these studies, and Duke DLAR for their assistance with mouse care. We acknowledge the Duke light microscopy core facility, the Duke Surgery Substrate Services Core research-support staff, and Translating Duke Health initiative for their support as well as StrideBio Inc., and Charles River laboratories for funding and technical assistance with NHP studies. Figures were created in part using BioRender.

## Author contributions

Conceptualization: T.J.G. and A.A. Methodology: T.J.G., T.J.S., D.K.O., L.P.H., L.B., A.L.R., G.W.D., M.K.E., and C.A.G. In vivo studies: T.J.G., L.B., K.S., M.M.F., A.L.R., M.S.M., R.M.C.R., and J.P. Investigation: T.J.G., L.B., K.S., M.M.F., A.L.R., M.S.M., and R.M.C.R. Visualization: T.J.G., L.B., and K.S. Funding acquisition: A.A., J.P., M.K.E., and C.A.G. Writing: T.J.G. and A.A.

## Competing interests

T.J.G, L.P.H., and A.A. have filed patent applications on the subject matter of this manuscript. A.A. is a co-founder at StrideBio and TorqueBio and serves on the advisory boards of Atsena Therapeutics, Isolere Bio, Mammoth Bio, Ring Therapeutics, and Kriya Therapeutics. C.A.G. is an advisor to Sarepta Therapeutics, Tune Therapeutics, Levo Therapeutics, and Iveric Bio and a co-founder of Tune Therapeutics, Element Genomics, and Locus Biosciences. R.M.C.R. and M.S.M. were employed at StideBio Inc., at this time. The remaining authors declare no competing interests.
