## [Peer Review File · Nature Communications]

Cross-species evolution of a highly potent AAV variant for therapeutic gene transfer and genome editingReviewer #1 (Remarks to the Author):

In this manuscript, Gonzalez et al. developed a cross-species evolution approach to evolve novel AAV capsid variants from AAV9 capsid from pig to mouse to NHP. The authors identified a novel variant AAV.cc47 that shows superior transduction and genome editing efficiency in mouse brain, heart, and TA muscle. In addition, via ICM and IT administration, AAV.cc47 outperforms AAV9 in brain transduction in NHP and piglet, respectively. The authors also demonstrated the potential of AAV.cc47 in treating rare diseases in the mouse models. These multi-faceted studies highlight the potential of cross-species evolution in generating novel AAV variants that can achieve highly efficient and comparable gene transduction among multiple species, suggesting that such a capsid engineering approach may serve as better predictive modeling to generate more efficient and clinically translatable AAV capsids for human gene therapy. Overall, the manuscript is well written with innovative science, data are clearly presented, supporting the conclusions drawn by the authors. However, the manuscript could be strengthened if the authors clarify the following issues.

Specific comments

1. The diversity of library is critical for capsid evolution. The author should indicate the size or diversity of each round library based on NGS analysis.
2. The total fluorescence and fluorescent intensity data presented in Fig. 2 and Fig. 6 are very impressive but less quantitatively informative. Did authors confirm the transduction and gene correction efficiencies by quantifying vector genome abundance, transgene mRNA or protein level?
3. In Fig. 1C, in addition to different species, it is not clear what the X axes represent?
4. For the data presented in Figure 4G and S6B, authors should discuss why gRNA but not Cas9 expression is boosted by AAV.cc47. Also, the data presented in Figure S6B is somewhat confusing, why does the mock group show high Cas9 expression?
5. On page 10, the end of the first paragraph, the authors stated: "This potentially corroborates the notion that AAV.cc47 has a post-entry advantage in transducing tissues compared to AAV9". What is the supporting data for this statement? Just a speculation? Authors should further clarify this.
6. Authors should expand description of the rationales for targeting VR-IV for saturated mutations from AAV structure biology points of view.
7. Authors should discuss why the systemic delivery route was not chosen for validating transduction efficiency of AAV.cc47, a novel capsid evolved by systemic delivery, in NHP and piglet.

Reviewer #2 (Remarks to the Author):

The study by Gonzalez et al is a tour de force in vector screening. I complement the leading authors on a large body of work, an interesting approach and solid data to support their findings. The manuscript applies a 3 species iterative selection/screening approach to one of the challenges in AAV vector development i.e. translation of performance across species. The screen enriched for AAV vectors that have increase vector genome DNA presence in a number of CNS regions. The library the authors generated for this effort was a focused saturation mutagenesis of variable region IV, believed to be one of the more important regions on the capsid for antibody binding and tropism. This screening effort yielded one clone of interest which the authors then further characterized in some depth; results show improvements in transduction efficiency of a number of therapeutically relevant target tissues *in vivo* |(mouse and nhp) versus a gold standard control AAV. These enhancement versus AAV9 overall seems modest, however in CSF injections in the NHP a more relevant improvement was noted in a limited study.

My overall assessment of the study is positive with the main strengths articulated above, however its impact and noteworthiness in its current form very much moderated.

The question to study species translation of AAV is interesting in the context of library screening. It appears however that the author's data is possibly more telling when analyzed much deeper than what is provided here. In fact, I would argue that a rigorous study of VR IV on this question would be very novel, informative, driving mechanistic hypotheses and possibly providing consistent design features to accelerate future AAV development. More focus however here is given to the characterization of a consensus sequence that survived the 3 species selection. Unfortunately, while the data shows some improvement it is modest and not sufficiently validated across species which is the primary goal of the manuscript. The NHP ICM CNS data is most compelling in terms impact for groups to possibly consider this new vector in my view, but needs further detailed characterization. In short, this is a tour de force, an interesting approach and question, but in my view focusing on an only modestly interesting novel capsid rather than the methodological unique approach misses the ball in terms of the novelty and impact. That said, the work deserves to be published, ideally with some of the points addressed below.

Open questions

- what have we learned about species translation? Is there additional analysis that can be performed for each of the individual species (e.g. representation of the library at each of the cycles, enrichment for certain motifs across species versus those that may be species specific, etc?)
- why the pig as a model for selection?
- How do you explain the bias for G at all positions in the 7-mer? Was the data normalized for the composition and representation in each of the infused libraries per cycle? The manuscript states 'slight' bias, however this seems to be an understatement and I suggest a more appropriate phrasing.
- How well was the tissue PCR represented in the eventual library that was subsequently infused in the next species selection (e.g. pig to mouse, and mouse to nhp)?
- Please provide relative to AAV1-9, parental library predominant sequence and other relevant comparisons ? Was it enriched at each cycle of the screen and if so by how much? Was there one species it was more beneficial vs others? Do the screening data align with the clonal data?
- Figure 3 states "AAV.cc47 outperforms AAV9 in Cre recombination in the Ai9 reporter mice at 810 all tested doses following intravenous (IV) administration." however the only quantitative analysis in the figure shows equal performance of both vectors. The histology shows perhaps in some panels a moderate improvement in some tissues at the highest dose. The supplemental fig 1 shows modest increases in TA and heart with limited statistical significance in some doses, but not all. Please revise. Given that the library was selected in the CNS, it seems logical to also have brain studied here with the more sensitive Cre system compared to mCherry in Fig2? Supplemental Fig 2 only seems to look at vector genomes, not expression (which why one would use the more sensitive Cre-based model?)
- the results in nhp CSF route of administration are some of the more compelling transduction efficiency improvements in the study. Given that the study was limited to n=2, can a more extensive dataset be presented from both animals and across more brain regions?

Reviewer #3 (Remarks to the Author):

In this paper, Trevor Gonzalez, Aravind Asokan, and colleagues present (i) a method of performing cross-species cycling during directed evolution of AAV capsid variants and (ii) a likely cross-species compatible variant (AAV.cc47) with improved transgene expression over AAV9. This paper addresses a critical need in the field of capsid engineering for better integration of different animal models in order to identify novel capsids with high therapeutic potential.

This study is quite compelling overall. The cross-species cycling strategy is a clever approach to find elusive cross-species compatible variants. The authors assess AAV.cc47

with a variety of transgenes and promoters, address both gene replacement and gene editing applications, and use two different mouse disease models. I was convinced by the data presented in this paper that AAV.cc47 facilitates enhanced transduction in mice, and the experiments in pigs and macaques suggest that this will be the case in these animal models as well. The data from pigs and macaques are rather thin, as the reporter studies use a small sample size and only qualitative data is presented. This work is nonetheless promising and justifies additional study of AAV.cc47 in primates; I would be especially interested in an assessment of AAV.cc47 following intravenous administration in NHPs. Future studies should also address toxicity concerns, particularly because it seems AAV.cc47 is at least as potent (if not more) as AAV9 in the liver. I recommend that this manuscript be accepted pending a satisfactory response to reviewer comments. Congratulations on your exciting findings.

Major comments:

- In line 164-165, it is claimed that AAV.cc47 transduces a greater number of neurons in a variety of regions of the mouse brain compared to AAV9. However, the referenced data in figure 2 only shows some qualitative fluorescence images (2C) and total mCherry fluorescence intensity across the sections (2D) and there is no assessment of transduction of neurons specifically. Please provide images showing localization of mCherry fluorescence with a neuron-specific marker (e.g. NeuN) and report percent transduced neurons for AAV9 and AAV.cc47.
- In figure 2 and figure 3/S1, please report the percentage of cells that are transduced instead of (or in addition to) total fluorescence intensity. For example, in figure 2 please report the ratio of mCherry+ cells to DAPI+ nuclei for each vector. This will provide a more robust assessment of transduction efficiency and align this data more closely to figure 4, where the percent of tdTomato+ cells is reported.
- In figure 6, please also assess the efficiency of Dmd gene editing by calculating the percent of Dmd transcripts with exon 23 deleted. This is probably best done with separate Taqman assays against the exon 22/24 junction and against an unaltered exon junction.
- In figure 5, please include the spinal cord and dorsal root ganglia along with a skeletal muscle if these tissues are available (TA is most consistent with earlier figures but other skeletal muscles would also work).
- My understanding of the description of the NHP reporter study in the methods section is that two macaques were injected with AAV9 and two were injected with AAV.cc47. If so, please include IHC images from all animals and additionally report vector genome biodistribution in the tissues of interest (brain, heart, liver, ideally also spinal cord, DRG, and skeletal muscle). If not and only one macaque was injected with each vector (as seems to be the case with the pig reporter study), please clarify this in the methods section.

Minor comments:

- In the introduction (lines 81-83), it is claimed that earlier studies yielding engineered capsids only performed iterative selections within a single animal model, rather than cycling between animal models as in this paper. However, some of the capsids presented in Tabebordbar et al (2021) were generated by cycling between mice and NHPs; specifically, MyoAAV 4A-4E were the result of a selection in NHPs on 120,000 variants enriched following a selection in mice.
- In the methods section, please describe the source of the parental library used to calculate fold enrichment of the evolved library after selection. Is the parental library sequenced from the rAAV library (i.e. what is injected into animals) or the plasmid library used to produce rAAV? The former will provide more accurate information about performance of each variant within the animal model, while the latter may be confounded by different rAAV production efficiencies of different plasmid variants.
- Please provide more detail on how vector genomes were recovered from tissue for sequencing. Why is the nucleic acid extracted from tissue being treated with DNase I (line 459)? Would this not destroy the delivered rAAV genomes that, once inside the cell, are no longer protected by the capsid?
- It seems like for production of rAAV libraries, full-length and functional AAV2 Rep is

provided in cis; i.e. AAV2 Rep is also encoded on the ITR-flanked genome packaged into capsids. Is this correct? If so, wouldn't the resulting rAAV libraries be replication competent if your animal model is also infected with a helper virus (this seems to be the highest concern for the NHPs)? I imagine that rAAV replication within the animal model(s) during your selection experiments could bias results- please address the validity of this concern.

- It is not clear what the different size circles mean in figure 1C.
- Please be explicit about the amino acid sequence of AAV.cc47 (line 139) as this was confusing- is it 452-GVSLGGG-458, or is that just a consensus sequence of the top variants?
- In figure 1F-G concerning vector production efficiency, is this comparison done with matched transgenes? In other words, are you comparing production of (for example) AAV9-eGFP to AAV.cc47-eGFP in a paired t-test, or are you comparing totally different sets of transgenes? Though I expect that the latter situation will be predictive of production efficiency to some degree, in my opinion the former comparison is more robust as it controls for any effect of the transgene on vector production. If possible, please do this analysis only on transgenes where you have data from both AAV9 and AAV.cc47. Please also note the type of statistical test performed in the figure legend.
- In figure 2B, you claim that transduction efficiency in the liver is the same for both vectors (line 161), though in figure 2D you show a statistically significant 2-fold increase in mCherry fluorescence from AAV.cc47 (line 162). Though the native fluorescence images do look qualitatively similar, I don't agree with the claim that transduction efficiency in the liver is the same between the two vectors given that you then go on to say there is a statistically significant difference in fluorescence intensity.
- In line 178 and elsewhere in the manuscript, please specify that you mean vector *genome* biodistribution.
- While the results from different vector designs for gene editing used in Figure 4 are illuminating, I'm not sure if this is a core component of the study. I suggest keeping design IV in the main figure (as this is the one that worked best) and moving the rest to the supplement. This will help make the figure less cluttered, as parts D, F, and G are a bit small and hard to read.
- Please add scale bars to all microscopy images.
- Please specify in the relevant figure legends which images of animal tissue are 50 μ M vibratome sections and which are 7 μ M cryostat sections- all I can find is a note in the legend for figure 2 that both types of sections are presented, but there is no indication which is which.
- In all figure legends please make sure you specify the age, dose, and transgene that is being delivered- this was mostly only an issue in the supplemental figure legends. Please also specify on relevant y axes "total mCherry fluorescence" (or whichever reporter you are looking at) rather than "total fluorescence".
- Please report the sex of the macaques used in the evolution and reporter experiments in the methods section.
- In line 302-309, it is suggested that the altered residues on AAV.cc47 could potentially aid in immune evasion, but you also note that the antigenic profile of AAV.cc47 is similar to that of AAV9. Please include this data if you wish to discuss this point. Additionally, even if the novel epitope itself is not immunogenic, if these pilot studies indicate that there is cross-reactivity between AAV9 antibodies and AAV.cc47, I'm not sure it's fair to suggest that AAV.cc47 might escape immune recognition without additional evidence.
- It is unclear what is meant in line 331-332 "Since our evolution approach did not select for capsids with increased transduction, but rather improved tissue tropism". Perhaps you mean that you selected for capsids with increased transduction in a particular tissue, rather than increased transduction overall?
- Regarding the statement in line 333-337 "we observed improved transgene expression, but similar [vector genome] biodistribution [...] the notion that transcript-based AAV library evolution strategies are more likely to yield variants with higher transduction efficiency is somewhat challenged by this observation"- it is indeed interesting that you see better expression of the transgene but not better delivery of the genome, perhaps this is due to a post-entry mechanism as you suggest, or perhaps it is a result of the nonlinear relationship between genome delivery, transcription, and

translation. I'm not sure if this observation has much bearing on the utility of transcript-based selection methods. Multiple groups have reported potent capsids identified using transcript-based methods (TRACER, DELIVER, TRADE). Though there are certainly other selection methods that can yield excellent results, as seen in this manuscript and in Cre-recombination based approaches (e.g. CREATE), transcript-based selection is a proven strategy as well.

Response to Reviewer #1:

“In this manuscript, Gonzalez et al. developed a cross-species evolution approach to evolve novel AAV capsid variants from AAV9 capsid from pig to mouse to NHP. The authors identified a novel variant AAV.cc47 that shows superior transduction and genome editing efficiency in mouse brain, heart, and TA muscle. In addition, via ICM and IT administration, AAV.cc47 outperforms AAV9 in brain transduction in NHP and piglet, respectively. The authors also demonstrated the potential of AAV.cc47 in treating rare diseases in the mouse models. These multi-faceted studies highlight the potential of cross-species evolution in generating novel AAV variants that can achieve highly efficient and comparable gene transduction among multiple species, suggesting that such a capsid engineering approach may serve as better predicative modeling to generate more efficient and clinically translatable AAV capsids for human gene therapy. Overall, the manuscript is well written with innovative science, data are clearly presented, supporting the conclusions drawn by the authors. However, the manuscript could be strengthened if the authors clarify the following issues.”

We would like to thank Reviewer #1 for the kind comments highlighting a well-written manuscript, innovative and clearly presented data. We have strengthened the manuscript as recommended and provide a point-by-point response below.

1. The diversity of library is critical for capsid evolution. The author should indicate the size or diversity of each round library based on NGS analysis.

We agree. This information is now included in **Supplemental Figure 1**. Both viral packaged genome diversity and amino acid representation is shown.

2. The total fluorescence and fluorescent intensity data presented in Fig. 2 and Fig. 6 are very impressive but less quantitatively informative. Did authors confirm the transduction and gene correction efficiencies by quantifying vector genome abundance, transgene mRNA or protein level?

The early reporter gene experiments shown in Fig. 2 were designed primarily to rapidly screen for clones with improved transduction profiles. Hence, only histological analysis was carried out with fixed tissues. It is important to note that fold increases in AAV transduction should be validated through multiple promoter-transgene cassettes and can vary with the type of genome packaged into the vector.

Correspondingly, we have quantified vector genomes and/or transgene mRNA as well as quantitative assessment of protein level quantifications for several other vectors throughout the paper including Cre recombinase and human acid-alpha glucosidase, which are showcased in **Figures 3D, 6B, 6C, and Supplemental Figures 4, 5D and 12**. Additionally, we have included DNA, RNA, and protein analysis with the mCherry transgene in NHPs, which can be found in **Supplemental Figure 7**.

Further, for CRISPR cassettes, vector genomes and transgene mRNA quantification for the genome editing studies are shown in **Figures 4 (Ai9 mice) and 6 (mdx mice) and Supplemental Figures 6 and 13**. Taken together, we feel that the totality of our data clearly attests to increased transduction efficiency of AAV.cc47 over AAV9.

3. In Fig. 1C, in addition to different species, it is not clear what the X axes represent?

The x-axis of each graph represents each AAV clone based on amino acid sequence. Each clone is randomly assigned a number during generation of the bubble plot in R and the x-axis is compressed to a maximum value of 1000. The figure legend has been updated to reflect the same.

4. For the data presented in Figure 4G and S6B, authors should discuss why gRNA but not Cas9 expression is boosted by AAV.cc47. Also, the data presented in Figure S6B is somewhat confusing, why does the mock group show high Cas9 expression?

Previous studies have shown that the CMV promoter is prone to silencing in muscle (e.g., Brooks et al., *J Gene Med*, 2004) and we have previously noted this to be the case in AAV-CRISPR muscle genome editing as well (Nelson et al., *Nat Med*, 2019). It is plausible that immune response to Cas9 in adult mice and promoter silencing could have contributed to this effect. We have observed low SaCas9 mRNA expression in both our adult Ai9 and mdx genome editing studies at 4 weeks post injection. The U6 promoter gRNAs don't appear to be subject to this at least at earlier time points reported here. We have addressed this aspect in the results section of the main text as well.

In regards to high SaCas9 mRNA expression in the mock control group, a delta delta Ct analysis was performed where the control group (mock) was set to 1 and the fold change from the treated groups plotted. It's not that the mock group is showing high SaCas9 expression, but that the treated groups are showing low to background levels of SaCas9 expression 4 weeks post injection with AAV9 and AAV.cc47 vectors. We have confirmed SaCas9 mRNA levels in both studies with two previously published primer pairs (Nelson et al., *Nat Med*, 2019).

5. On page 10, the end of the first paragraph, the authors stated: "This potentially corroborates the notion that AAV.cc47 has a post-entry advantage in transducing tissues compared to AAV9". What is the supporting data for this statement? Just a speculation? Authors should further clarify this.

We made this statement based on the equal copy numbers recovered from different tissues regardless of transgene after dosing AAV.cc47 or AAV9 vectors. In other words, no significant differences in uptake/biodistribution are noted. If acceptable, we have now modified the main text sentence to read "Based on these observations, it is tempting to speculate that AAV.cc47 may not have a binding/uptake advantage in tissues, but may benefit from a potential post-entry event compared to AAV9."

6. Authors should expand description of the rationales for targeting VR-IV for saturated mutations from AAV structure biology points of view.

The following text has now been added to the main text. "Briefly, binding interactions of galactosylated glycans and AAVR with the AAV9 capsid are known to occur around the 3-fold axis. In addition to the functional attributes outlined above, previous studies by our lab (Tse et al., *PNAS* 2017; Havlik et al., *J Virol*, 2020) and the Gradinaru lab (Goersten et al., *Nat Neurosci*, 2022) have demonstrated the importance of VR-IV in evolving novel AAV capsids with improved functionality. Moreover, the VR-IV loop is less permissive towards peptide insert approaches compared to VR-VIII. Taken together, VR-IV is an important surface region for incorporation into AAV capsid libraries."

7. Authors should discuss why the systemic delivery route was not chosen for validating transduction efficiency of AAV.cc47, a novel capsid evolved by systemic delivery, in NHP and piglet.

At this writing, we were unable to carry out IV dosing due to dosing and immunological considerations. Multiple studies by other groups have demonstrated that reporter gene expression in NHPs and pigs can lead to significant transgene-specific immune toxicity. Since our goal was to prioritize validation of increased potency in multiple species rather than biodistribution, we chose to showcase transgene expression following a lower dose and CSF route at an early time point – as evident from our data in **Figure 5** and **Supplementary Figures 7-10 (NHPs) and 11C-D (pigs)**, this strategy confirmed the improved potency of AAV.cc47 in CNS as well as systemic organs such as the heart and liver. We are currently optimizing immunosuppression regimens and alternative cassettes for follow up systemic dosing studies in NHP and pig cohorts.

Response to Reviewer #2

“The study by Gonzalez et al is a tour de force in vector screening. I complement the leading authors on a large body of work, an interesting approach and solid data to support their findings. The manuscript applies a 3 species iterative selection/screening approach to one of the challenges in AAV vector development i.e. translation of performance across species. The screen enriched for AAV vectors that have increase vector genome DNA presence in a number of CNS regions. The library the authors generated for this effort was a focused saturation mutagenesis of variable region IV, believed to be one of the more important regions on the capsid for antibody binding and tropism. This screening effort yielded one clone of interest which the authors then further characterized in some depth; results show improvements in transduction efficiency of a number of therapeutically relevant target tissues in vivo (mouse and nhp) versus a gold standard control AAV. These enhancement versus AAV9 overall seems modest, however in CSF injections in the NHP a more relevant improvement was noted in a limited study.

My overall assessment of the study is positive with the main strengths articulated above, however its impact and noteworthiness in its current form very much moderated. The question to study species translation of AAV is interesting in the context of library screening. It appears however that the author's data is possibly more telling when analyzed much deeper than what is provided here. In fact, I would argue that a rigorous study of VR IV on this question would be very novel, informative, driving mechanistic hypotheses and possibly providing consistent design features to accelerate future AAV development. More focus however here is given to the characterization of a consensus sequence that survived the 3 species selection. Unfortunately, while the data shows some improvement it is modest and not sufficiently validated across species which is the primary goal of the manuscript. The NHP ICM CNS data is most compelling in terms impact for groups to possibly consider this new vector in my view, but needs further detailed characterization. In short, this is a tour de force, an interesting approach and question, but in my view focusing on an only modestly interesting novel capsid rather than the methodological unique approach misses the ball in terms of the novelty and impact. That said, the work deserves to be published, ideally with some of the points addressed below.”

“My overall assessment of the study is positive with the main strengths articulated above, however its impact and noteworthiness in its current form very much moderated. “

First, we really appreciate Reviewer #2 for their comments characterizing our study as a “tour-de-force” in vector screening, having solid data supporting our findings and deserving of publication. We have responded in a point-by-point manner that addresses the concerns above.

We respectfully disagree with this specific statement above. About a third of the AAV gene therapy trials to date are facing clinical holds due to issues with serious adverse events. Relatively low transduction efficiency of natural AAV serotypes has necessitated high doses leading to toxicity. There is an unmet need for AAV vectors in the clinic with improved potency that can enable a broader therapeutic window. In addition to our screening effort in multiple species for the first time, we would like to reiterate that our study exhaustively evaluates multiple transgenes, multiple mouse models and multiple species to validate this new capsid. The true impact of our study as well others can only be vetted in the clinic. In the meantime, we feel strongly that the availability of improved screening strategies and capsids with truly differentiated and meaningful properties remains noteworthy and of high significance to the Gene Therapy community.

“The question to study species translation of AAV is interesting in the context of library screening. It appears however that the author's data is possibly more telling when analyzed much deeper than what is provided here.”

We agree. Further mechanistic analysis of results from cross-species evolution (e.g., order of species or different AAV libraries) is warranted. Mechanistic studies as well as in-depth analysis of the library approach is currently underway, but outside the scope of the current study. The current study is specifically focused on developing and validating a newly evolved capsid displaying higher potency.

“In fact, I would argue that a rigorous study of VR IV on this question would be very novel, informative, driving mechanistic hypotheses and possibly providing consistent design features to accelerate future AAV development. More focus however here is given to the characterization of a consensus sequence that survived the 3 species selection.”

In addition to studies published with the late Dr. Agbandje-McKenna's group on the structural importance of VR-IV (e.g., Pulicherla et al., Mol Ther, 2011; Emmanuel et al., J Virol, 2022), we have recently published three studies highlighting the importance of VR-IV in AAV biology as well as AAV library engineering and screening (Tse et al., PNAS, 2017; Havlik et al., J Virol, 2020; Havlik et al., JV, 2021). The focus on the novel consensus sequence arising from three completely different species that are pertinent preclinical models is the central focus of this study. We have identified a novel AAV variant with improved potency and provided substantial evidence to corroborate the same.

“Unfortunately, while the data shows some improvement it is modest and not sufficiently validated across species which is the primary goal of the manuscript. The NHP ICM CNS data is most compelling in terms impact for groups to possibly consider this new vector in my view, but needs further detailed characterization.”

It is important to understand that performance of AAV vectors is not just determined by the capsid alone. As the reviewer is aware, choice of promoter, transgene, other regulatory elements in the genome and preclinical models – all impact transgene expression. Our study unequivocally

demonstrates improved potency of AAV.cc47 benchmarked over AAV9 through reporter gene expression, Cre recombination, genome editing in two different mouse models and therapeutic gene expression (Pompe model). We validated improved AAV.cc47 vector mediated transgene expression using single-stranded and self-complementary genomes, 4 different promoters and 8 different transgene cassettes in this study across 3 animal species. While the transduction efficiencies might vary with these different conditions, our results unequivocally support the improved phenotype presented by AAV.cc47.

“In short, this is a tour de force, an interesting approach and question, but in my view focusing on an only modestly interesting novel capsid rather than the methodological unique approach misses the ball in terms of the novelty and impact. That said, the work deserves to be published, ideally with some of the points addressed below.”

We really appreciate Reviewer #2 for their comments characterizing our study as a “tour-de-force” in vector screening, having solid data supporting our findings and deserving of publication. Please see our comment above regarding impact on the field.

1. what have we learned about species translation? Is there additional analysis that can be performed for each of the individual species (e.g. representation of the library at each of the cycles, enrichment for certain motifs across species versus those that may be species specific, etc?)

We would like to thank the reviewer for this suggestion. We have now included additional analyses pertaining to NGS data for each species in **Supplemental Figure 1**. In particular, we have now included consensus motif analysis for each species in the **Supplementary Figure 1**. Specific and noteworthy observations are mentioned below and the discussion section has been updated.

“The cross-species evolution approach revealed several interesting trends. At a macro level, AAV library cycling leads to decreased sequence diversity in each cycling step. Whether this phenomenon arises from the number of cycles alone or is also influenced by a specific animal model is unclear. It is interesting to note that AAV.cc47 or related variants were not enriched as top leads when we started library cycling *in vitro*, *in vivo* in mice rather than pigs or when cycling in NHPs alone (data not shown). Nonetheless, it is interesting to note that the percent representation of conserved amino acid residues for each position in VR-IV increases particularly after cycling in NHPs. In particular, we observed significant enrichment of neutral amino acid residues with no to small side chains (esp. glycine, alanine, valine) in VR-IV as we cycled across each species. The final NHP cycling step markedly enriched Gly, Ser and Leu residues within VR-IV. Amongst these, the Gly residues were preferably enriched in multiple species. Novel receptor or host factor usage and other functional ramifications of these structural changes are as yet unknown. It is however pertinent to mention that AAV.cc47 is still AAVR-dependent for cellular infection (similar to AAV9; **Supplementary Figure 1**).”

2. why the pig as a model for selection?

Pigs have emerged as a critical preclinical model in evaluating AAV gene therapies as well as other therapeutic modalities across multiple organ systems (Lunney et al., *Sci Transl Med* 2021; Hinderer et al. *Hum Gene Ther* 2018; Unger et al., 2017). Pigs are also particularly relevant in and commonly

utilized in cardiac-failure and other AAV pre-clinical gene therapy studies (Weber et al., Gene Therapy 2013; Raake et al., Gene Therapy 2007; Mussolino et al., Gene Therapy 2011). Moreover, in addition to logistical (emerging shortage of macaques) and cost related advantages, pigs have also remained a staple in the human organ transplantation field over the last decade (e.g., Allison, S., Nat Rev Neph 2022).

3. How do you explain the bias for G at all positions in the 7-mer? Was the data normalized for the composition and representation in each of the infused libraries per cycle? The manuscript states 'slight' bias, however this seems to be an understatement and I suggest a more appropriate phrasing.

As outlined earlier, AAV.cc47 or related variants were not enriched as top leads when we started library cycling *in vitro*, *in vivo* in mice rather than pigs or when cycling in NHPs alone (data not shown). Over the past five years of working with different AAV libraries, based on a range of different AAV serotypes and engineering different regions on the capsid surface, we have not observed this phenomenon. Hence, we strongly believe glycine preference here is driven by host-specific selection pressure. The perceived bias can possibly be explained by the fact that Glycine residues present less structural constraint for key capsid protein interactions as well as capsid-host interactions, due to lack of side chains. Whether this is a loss of function (lack of recognition by host restriction factor) or gain of function (recognition of a cross-species conserved host factor) remains to be seen. At this writing, our biodistribution data suggests that binding or uptake in tissues is not improved; however, increased transcript levels and transgene expression are noted regardless of the vector genome delivered. The latter observation would suggest a post-entry mechanism is likely at play. It is also important to note that each position is not a Gly – in fact other residues such Val, Ser, Leu are also significantly enriched in the first half of the evolved VR-IV epitope.

The data is normalized. Each evolved/enriched variant is normalized to the representation in the parent library. Additional clarification is presented in a response below. Also please note, we have updated the main text from a “slight bias” to “bias” in the main text.

4. How well was the tissue PCR represented in the eventual library that was subsequently infused in the next species selection (e.g. pig to mouse, and mouse to nhp)?

All PCR products were cloned as is into the subsequent library allowing natural selection mediated enrichment. We only carry out NGS analysis of viral libraries (parental and evolved), since this is the most appropriate metric for corroborating yield (titer) and enrichment due to species-specific selection pressure *in vivo*.

5. Please provide relative to AAV1-9, parental library predominant sequence and other relevant comparisons ? Was it enriched at each cycle of the screen and if so by how much? Was there one species it was more beneficial vs others? Do the screening data align with the clonal data?

```

AAV1      QYLYYLNRTQ-NQSGSAQNKDLLFSRGS- 27
AAV2      QYLYYLSRTN-TPSGTTTQSRQLQFSQAG- 27
AAV3      QYLYYLNRTQGTTSGTTNQSRLLSQA-- 27
AAV4      QYLWGLQSTTIGTTLNAGTATTNFTKL-- 27
AAV5      QYLYRFVSTN-NTGGVQFNKLAGRYAN- 27
AAV6      QYLYYLNRTQ-NQSGSAQNKDLLFSRGS- 27
AAV7      QYLYYLARTQSNPGGTAGMRELQFYQG-- 27
AAV8      QYLYYLSRTQ-TTGGTANTQTLGFSQGG- 27
AAV9      QYLYYLSKTINGSGQ--NQQLKFSVAGP 27
AAVrh10   QYLYYLSRTQ-STGGTAGTQQLLFSQAG- 27
AAV.cc47  QYLYYLSKTIQVSLG--GGQTLKFSVAGP 27
          ****. . *

```

Capsid	Sequence	Parental %	Evolved %	X-Enrichment	Animal
AAV.cc47	GVSLGGG	0.00089	0.0011	1	pig
AAV9	NGSGQNQ	0.99088	4.2210	4	
AAV.cc47	GVSLGGG	0.00089	0.1339	150	mouse
AAV9	NGSGQNQ	0.99088	20.9090	21	
AAV.cc47	GVSLGGG	0.00089	61.5220	68934	nhp
AAV9	NGSGQNQ	0.99088	14.5160	15	

Sequence alignments with different AAV serotypes are shown above. Some serotypes including AAV5, 7, 8 and rh.10 do show the presence of Gly residues, but always at the same position. Specifically, AAV.cc47 appears to prefer Gly residues in the latter half of VR-IV. The specific enrichment parameters are outlined in the Table above. This information has also been included in **Supplementary Table 2**. Please see our earlier response to Q1 on species preference and related explanations. Lastly, the enriched DNA sequences should be expected to show greater diversity (degenerate codons), while we observe greater conservation at the amino acid level upon cycling.

6. Figure 3 states "AAV.cc47 outperforms AAV9 in Cre recombination in the Ai9 reporter mice at 810 all tested doses following intravenous (IV) administration." however the only quantitative analysis in the figure shows equal performance of both vectors. The histology shows perhaps in some panels a moderate improvement in some tissues at the highest dose. The supplemental fig 1 shows modest increases in TA and heart with limited statistical significance in some doses, but not all. Please revise.

We thank the reviewer for this suggestion. We have now re-analyzed the data and provide additional and definitive quantitative analysis in the form of "% tdtomato+ cells normalized to dapi+ nuclei" reflecting Cre-mediated recombination efficiency heart, skeletal muscle, and liver (Please see new **Figure 3**). This detailed analysis affirms the improved gene transfer efficiency. It is critical that the histology data be considered in totality of the multiple doses evaluated, since one would expect saturation of Cre-mediated recombination at higher doses. Overall, this data corroborates that AAV.cc47 is more potent than AAV9.

Given that the library was selected in the CNS, it seems logical to also have brain studied here with the more sensitive Cre system compared to mCherry in Fig2? Supplemental Fig 2 only seems to look at vector genomes, not expression (which why one would use the more sensitive Cre-based model?)

We have now included CNS expression data for AAV Cre-mediated recombination in **Supplementary Figure 5**. Histology, Neuronal marker (NeuN) co-immunostaining, quantitation and vector genome biodistribution in the CNS are included in the revision. It is noteworthy to mention that the CMV promoter used in the current study to drive Cre expression is prone to silencing in the CNS (e.g., Gray et al., Hum Gene Ther, 2011) and this transgene cassette is not ideally suited for CNS Cre recombination.

7. the results in nhp CSF route of administration are some of the more compelling transduction efficiency improvements in the study. Given that the study was limited to n=2, can a more extensive dataset be presented from both animals and across more brain regions?

We have now carried out an extensive analysis of the tissues obtained from the treated NHP cohorts and include new **Supplementary Figures 7-10**. In particular, we provide new data analysis showcasing vector genome biodistribution, mRNA expression levels, as well as mCherry expression in different regions of the brain (premotor cortex, thalamus, dentate nucleus, cerebellum) as well as heart and liver.

Response to reviewer #3

“In this paper, Trevor Gonzalez, Aravind Asokan, and colleagues present (i) a method of performing cross-species cycling during directed evolution of AAV capsid variants and (ii) a likely cross-species compatible variant (AAV.cc47) with improved transgene expression over AAV9. This paper addresses a critical need in the field of capsid engineering for better integration of different animal models in order to identify novel capsids with high therapeutic potential.

This study is quite compelling overall. The cross-species cycling strategy is a clever approach to find elusive cross-species compatible variants. The authors assess AAV.cc47 with a variety of transgenes and promoters, address both gene replacement and gene editing applications, and use two different mouse disease models. I was convinced by the data presented in this paper that AAV.cc47 facilitates enhanced transduction in mice, and the experiments in pigs and macaques suggest that this will be the case in these animal models as well. The data from pigs and macaques are rather thin, as the reporter studies use a small sample size and only qualitative data is presented. This work is nonetheless promising and justifies additional study of AAV.cc47 in primates; I would be especially interested in an assessment of AAV.cc47 following intravenous administration in NHPs. Future studies should also address toxicity concerns, particularly because it seems AAV.cc47 is at least as potent (if not more) as AAV9 in the liver. I recommend that this manuscript be accepted pending a satisfactory response to reviewer comments. Congratulations on your exciting findings.”

We would like to thank Reviewer #3 for their comments stating the study as compelling, clever and exciting as well as recommending the manuscript for publication pending additional improvements. We have now taken these suggestions into consideration and present a significantly improved manuscript.

“I would be especially interested in an assessment of AAV.cc47 following intravenous administration in NHPs. Future studies should also address toxicity concerns, particularly because it seems AAV.cc47 is at least as potent (if not more) as AAV9 in the liver.”

We agree. These are high priority studies prior to advancing AAV.cc47 for clinical evaluation.

1. In line 164-165, it is claimed that AAV.cc47 transduces a greater number of neurons in a variety of regions of the mouse brain compared to AAV9. However, the referenced data in figure 2 only shows some qualitative fluorescence images (2C) and total mCherry fluorescence intensity across the sections (2D) and there is no assessment of transduction of neurons specifically. Please provide images showing localization of mCherry fluorescence with a neuron-specific marker (e.g. NeuN) and report percent transduced neurons for AAV9 and AAV.cc47.

We apologize for this oversight. We would like to point out that AAV.cc47 transduces many different brain cell types, not just neurons. This is corroborated by the use of ubiquitous promoters driving different transgenes. Nevertheless, we have now carried out co-immunofluorescence staining studies with a neuronal marker (NeuN) as requested and have now updated **Figure 2** with quantitative cell count data showing 3-4 fold higher percentage of neurons transduced in the cerebellum, hippocampus and cortical regions. Representative fluorescent images from the co-staining study have now been included in **Supplementary Figure 2** as well.

2. In figure 2 and figure 3/S1, please report the percentage of cells that are transduced instead of (or in addition to) total fluorescence intensity. For example, in figure 2 please report the ratio of mCherry+ cells to DAPI+ nuclei for each vector. This will provide a more robust assessment of transduction efficiency and align this data more closely to figure 4, where the percent of tdTomato+ cells is reported.

We would like to thank the reviewer for this suggestion. We have now carried this analysis for liver, heart and skeletal muscle tissues in **Figure 3** (AAV Cre recombination study). With regard to Figure 2, we would like to note that the mCherry study was part of an initial screening effort to assess transduction efficiencies from a large number of evolved variants. Hence the data from these fixed tissues is only semi-quantitative and corroborated by our quantitative analysis with multiple other transgenes in the Cre, GAA, Ai9 and mdx editing studies in mouse models.

3. In figure 6, please also assess the efficiency of Dmd gene editing by calculating the percent of Dmd transcripts with exon 23 deleted. This is probably best done with separate Taqman assays against the exon 22/24 junction and against an unaltered exon junction.

Thank you for this excellent suggestion. We have now carried out additional studies to quantify genome editing efficiency in the mdx mouse model. As shown in newly included **Figure 6H**, we have now carried out Taqman qPCR to demonstrate the percentage of Exon 23 deleted transcripts. The data clearly shows

a significant 3-5 fold improvement in editing efficiency using AAV.cc47 compared to AAV9 in heart and tibialis anterior skeletal muscle. These new data have now been included in the main text.

4. In figure 5, please include the spinal cord and dorsal root ganglia along with a skeletal muscle if these tissues are available (TA is most consistent with earlier figures but other skeletal muscles would also work).

We have included lumbar, thoracic, and cervical spinal cord histology in **Supplementary Figure 10**. Unfortunately, skeletal muscle was not harvested from the NHPs due to the low dose and route of administration utilized in this study, although we would predict improved transduction based on the cardiac dataset.

5. My understanding of the description of the NHP reporter study in the methods section is that two macaques were injected with AAV9 and two were injected with AAV.cc47. If so, please include IHC images from all animals and additionally report vector genome biodistribution in the tissues of interest (brain, heart, liver, ideally also spinal cord, DRG, and skeletal muscle). If not and only one macaque was injected with each vector (as seems to be the case with the pig reporter study), please clarify this in the methods section.

Images from both NHPs have been included in **Supplementary Figures 8 – 10**, which include brain, spinal cord, heart and muscle. Moreover, we also provide additional quantitative analyses for vector genome biodistribution, mCherry mRNA levels and mCherry protein levels in **Supplementary Figure 7**.

Minor comments:

1. In the introduction (lines 81-83), it is claimed that earlier studies yielding engineered capsids only performed iterative selections within a single animal model, rather than cycling between animal models as in this paper. However, some of the capsids presented in Tabebordbar et al (2021) were generated by cycling between mice and NHPs; specifically, MyoAAV 4A-4E were the result of a selection in NHPs on 120,000 variants enriched following a selection in mice.

We apologize for this oversight. We have now removed this claim and edited these lines to read “A notable drawback that has been reported is that it is plausible that directed evolution of AAV libraries in a single animal model can yield variants specific for those species, as exemplified in case of AAV-PHP.B (22), which utilizes the C57/B6 mouse strain-specific receptor, Ly6A (23).”

2. In the methods section, please describe the source of the parental library used to calculate fold enrichment of the evolved library after selection. Is the parental library sequenced from the rAAV library (i.e. what is injected into animals) or the plasmid library used to produce rAAV? The former will provide more accurate information about performance of each variant within the animal model, while the latter may be confounded by different rAAV production efficiencies of different plasmid variants.

The parental library and evolved libraries were sequenced by NGS using the AAV libraries and not plasmid. We have now specified this aspect in the methods section. Further, we also include new

analysis of the library across different species, diversity and relative enrichment parameters in **Supplementary Figure 1**.

3. Please provide more detail on how vector genomes were recovered from tissue for sequencing. Why is the nucleic acid extracted from tissue being treated with DNase I (line 459)? Would this not destroy the delivered rAAV genomes that, once inside the cell, are no longer protected by the capsid?

We apologize for the confusion, the viral genomes extracted from tissue are not treated with DNase-I. Our intent was to point out the purified viral libraries were pretreated with DNase-I PRIOR to extraction of the viral genomes for subsequent NGS preparation. We have now updated the main text to reflect this. "Sequence diversity of the parental and evolved AAV9 capsid libraries were assessed by using the viral library preparations. The evolved library sequences were generated following PCR of DNA amplified tissue followed by plasmid cloning and viral production. Both the parental and evolved viral libraries after production were DNase-I treated prior to extraction of viral genomes from the capsids and subsequent addition of Illumina adaptor sequences via PCR."

4. It seems like for production of rAAV libraries, full-length and functional AAV2 Rep is provided in cis; i.e. AAV2 Rep is also encoded on the ITR-flanked genome packaged into capsids. Is this correct? If so, wouldn't the resulting rAAV libraries be replication competent if your animal model is also infected with a helper virus (this seems to be the highest concern for the NHPs)? I imagine that rAAV replication within the animal model(s) during your selection experiments could bias results- please address the validity of this concern.

We have carried out previous evolution studies with wild type AAV libraries and indeed one should expect replication in the presence of helper virus co-infection (e.g., Tse et al., PNAS 2017; Havlik et al., J Virol, 2021). It should however be noted that NHPs and pigs are typically screened for Herpes. Further, Adenoviral infections are typically restricted to upper respiratory or GI tracts and our studies were carried out in healthy animals. While one cannot entirely eliminate the possibility of an infection, since our route of library cycling is intravenous, we do not expect any appreciable replication in these epithelial tissues. It should also be noted that any natural co-infection by a helper virus would have resulted in replication (and enrichment) of AAV variants in certain tissues over others. We did not observe such in our evolution study and as shown, our efforts yielded a pan-tropic AAV variant.

5. It is not clear what the different size circles mean in figure 1C.

Each bubble in the plots in **Figure 1** correspond to an individual amino acid sequence from the libraries and the size of the bubbles correspond to the fold enrichment of that sequence from the parental library in the evolved library. This has been specified in the figure legend.

6. Please be explicit about the amino acid sequence of AAV.cc47 (line 139) as this was confusing- is it 452-GVSLGGG-458, or is that just a consensus sequence of the top variants?

The amino acid sequence for AAV.cc47 is 452-GVSLGGG-458. We have also provided consensus motif analysis for each species in **Supplementary Figure 1**.

7. In figure 1F-G concerning vector production efficiency, is this comparison done with matched transgenes? In other words, are you comparing production of (for example) AAV9-eGFP to AAV.cc47-eGFP in a paired t-test, or are you comparing totally different sets of transgenes? Though I expect that the latter situation will be predictive of production efficiency to some degree, in my opinion the former comparison is more robust as it controls for any effect of the transgene on vector production. If possible, please do this analysis only on transgenes where you have data from both AAV9 and AAV.cc47. Please also note the type of statistical test performed in the figure legend.

The goal of this dataset is to demonstrate that AAV.cc47 can be produced as effectively of AAV9 regardless of the transgene packaged into these capsids. In addition, we show that these vectors can be generated in both adherent and suspension 293 producer cells without any changes to yield/vector genome titers. In total, this data is most powerful if represented as a composite of AAV yield independent of the transgene used. We have now included separate symbols for each transgene and carried out a paired t-test to corroborate our observation that AAV.cc47 is amenable to manufacturing scale up similar to AAV9 vectors.

8. In figure 2B, you claim that transduction efficiency in the liver is the same for both vectors (line 161), though in figure 2D you show a statistically significant 2-fold increase in mCherry fluorescence from AAV.cc47 (line 162). Though the native fluorescence images do look qualitatively similar, I don't agree with the claim that transduction efficiency in the liver is the same between the two vectors given that you then go on to say there is a statistically significant difference in fluorescence intensity.

We have now revised this to read "a modest, yet statistically significant increase in liver transduction."

9. In line 178 and elsewhere in the manuscript, please specify that you mean vector *genome* biodistribution.

Thanks for pointing out this omission. We have now updated "vector biodistribution" to "vector genome biodistribution" in the appropriate places.

10. While the results from different vector designs for gene editing used in Figure 4 are illuminating, I'm not sure if this is a core component of the study. I suggest keeping design IV in the main figure (as this is the one that worked best) and moving the rest to the supplement. This will help make the figure less cluttered, as parts D, F, and G are a bit small and hard to read.

We would like to thank the reviewer for this suggestion. We have now modified **Figure 4** to reflect only the self-complementary (optimized) guide RNA cassette. Data pertaining to the remaining cassettes is now shown in **Supplementary Figure 6**.

11. Please add scale bars to all microscopy images.

Thanks for pointing out this omission. We have added scale bars to each image.

12. Please specify in the relevant figure legends which images of animal tissue are 50 μ M vibratome

sections and which are 7 uM cryostat sections- all I can find is a note in the legend for figure 2 that both types of sections are presented, but there is no indication which is which.

We apologize for this oversight. The figure legends have been updated to include this information where appropriate.

13. In all figure legends please make sure you specify the age, dose, and transgene that is being delivered- this was mostly only an issue in the supplemental figure legends. Please also specify on relevant y axes “total mCherry fluorescence” (or whichever reporter you are looking at) rather than “total fluorescence”.

The y axes and figure legends in **Figure 2 and 6** as well as all **Supplementary Figures** have now been updated to reflect the reporter fluorescence intensity measured. We have also made sure to include the age, dose, and transgene used in each figure legend.

14. Please report the sex of the macaques used in the evolution and reporter experiments in the methods section.

All macaques utilized were 2-3 year old females (vendor availability). This has now been updated in the methods section.

15. In line 302-309, it is suggested that the altered residues on AAV.cc47 could potentially aid in immune evasion, but you also note that the antigenic profile of AAV.cc47 is similar to that of AAV9. Please include this data if you wish to discuss this point. Additionally, even if the novel epitope itself is not immunogenic, if these pilot studies indicate that there is cross-reactivity between AAV9 antibodies and AAV.cc47, I’m not sure it’s fair to suggest that AAV.cc47 might escape immune recognition without additional evidence.

We apologize for this error. Assessment of the immunogenicity of AAV.cc47 compared to AAV9 will require a large NHP cohort in a separately designed study. We have however evaluated the ability of AAV.cc47 to evade neutralizing antibodies (NAbs) from pig, NHP and human antisera. Our studies indicate that AAV.cc47 is neutralized similar to AAV9 and corroborates earlier observations (Tse et al., PNAS 2017) that the newly evolved antigenic epitope in VR-IV alone is not sufficient for NAb evasion. We have now included this data in **Supplementary Figures 14A-C** and updated the main text accordingly.

16. It is unclear what is meant in line 331-332 “Since our evolution approach did not select for capsids with increased transduction, but rather improved tissue tropism”. Perhaps you mean that you selected for capsids with increased transduction in a particular tissue, rather than increased transduction overall?

Thanks for catching this error. The statement should read “Since our evolution approach did not select for capsids with improved tropism, but rather enhanced transduction efficiency”. We have modified the main text to reflect this correction.

17. Regarding the statement in line 333-337 “we observed improved transgene expression, but similar [vector genome] biodistribution [...] the notion that transcript-based AAV library evolution strategies are more likely to yield variants with higher transduction efficiency is somewhat challenged by this observation” - it is indeed interesting that you see better expression of the transgene but not better delivery of the genome, perhaps this is due to a post-entry mechanism as you suggest, or perhaps it is a result of the nonlinear relationship between genome delivery, transcription, and translation. I'm not sure if this observation has much bearing on the utility of transcript-based selection methods. Multiple groups have reported potent capsids identified using transcript-based methods (TRACER, DELIVER, TRADE). Though there are certainly other selection methods that can yield excellent results, as seen in this manuscript and in Cre-recombination based approaches (e.g. CREATE), transcript-based selection is a proven strategy as well.

We agree and have removed this sentence.

Reviewer #1 (Remarks to the Author):

in this revised version of the manuscript, the authors addressed the reviewer's comments adequately.

Reviewer #2 (Remarks to the Author):

This revision address all the major concerns raised by this reviewer.

Reviewer #3 (Remarks to the Author):

I greatly appreciate the authors' thoughtful and detailed response to reviewer comments. The addition of percent neuronal transduction quantification supports the overall findings of the paper, and the representative images in figure S2 look nice. The quantification of Dmd exon 23 deletion also strengthens the paper, it is good to see an improvement in both cardiac and skeletal muscle tissue on this front. Thank you for including IHC images from the DRG and other tissues in figure S10, they are very informative. It's hard to say for sure with just two pictures, but it seems plausible that AAV.cc47 transduces the DRG more than AAV9. It's probably out of scope for this study but I encourage the authors to assess DRG transduction and toxicity in future studies of AAV.cc47 in NHPs. I am satisfied by the edits addressing all of my minor comments. It was a pleasure reading this manuscript.

Response to Reviewer #1:

“In this revised version of the manuscript, the authors addressed the reviewer's comments adequately.”

We would like to thank Reviewer #1 for their time and recommending this manuscript be published.

Response to Reviewer #2

“This revision address all the major concerns raised by this reviewer.”

We would like to thank Reviewer #2 for their time and recommending this manuscript be published.

Response to reviewer #3

“I greatly appreciate the authors' thoughtful and detailed response to reviewer comments. The addition of percent neuronal transduction quantification supports the overall findings of the paper, and the representative images in figure S2 look nice. The quantification of Dmd exon 23 deletion also strengthens the paper, it is good to see an improvement in both cardiac and skeletal muscle tissue on this front. Thank you for including IHC images from the DRG and other tissues in figure S10, they are very informative. It's hard to say for sure with just two pictures, but it seems plausible that AAV.cc47 transduces the DRG more than AAV9. It's probably out of scope for this study but I encourage the authors to assess DRG transduction and toxicity in future studies of AAV.cc47 in NHPs. I am satisfied by the edits addressing all of my minor comments. It was a pleasure reading this manuscript.”

We would like to thank Reviewer #3 for their time and recommending this manuscript be published. We are happy the additional requested data supports the overall findings of this manuscript.